# TROP2 Represents a Negative Prognostic Factor in Colorectal Adenocarcinoma and Its Expression Is Associated with Features of Epithelial–Mesenchymal Transition and Invasiveness

**DOI:** 10.3390/cancers14174137

**Published:** 2022-08-26

**Authors:** Jiří Švec, Monika Šťastná, Lucie Janečková, Dušan Hrčkulák, Martina Vojtěchová, Jakub Onhajzer, Vítězslav Kříž, Kateřina Galušková, Eva Šloncová, Jan Kubovčiak, Lucie Pfeiferová, Jan Hrudka, Radoslav Matěj, Petr Waldauf, Lukáš Havlůj, Michal Kolář, Vladimír Kořínek

**Affiliations:** 1Laboratory of Cell and Developmental Biology, Institute of Molecular Genetics of the Czech Academy of Sciences, Vídeňská 1083, 142 20 Prague, Czech Republic; 2Department of Oncology, Third Faculty of Medicine, Charles University, University Hospital Kralovské Vinohrady, Šrobárova 1150/50, 100 34 Prague, Czech Republic; 3Laboratory of Genomics and Bioinformatics, Institute of Molecular Genetics of the Czech Academy of Sciences, Vídeňská 1083, 142 20 Prague, Czech Republic; 4Department of Informatics and Chemistry, Faculty of Chemical Technology, University of Chemistry and Technology Prague, 166 28 Prague, Czech Republic; 5Department of Pathology, Third Faculty of Medicine, Charles University, University Hospital Kralovské Vinohrady, Šrobárova 1150/50, 100 34 Prague, Czech Republic; 6Department of Pathology and Molecular Medicine, Third Medical Faculty, Charles University, Thomayer University Hospital, Ruská 87, 100 00 Praha, Czech Republic; 7Department of Anaesthesia and Intensive Care Medicine, Third Faculty of Medicine, Charles University, University Hospital Kralovské Vinohrady, Šrobárova 1150/50, 100 34 Prague, Czech Republic; 8Department of General Surgery, Third Faculty of Medicine, Charles University, University Hospital Kralovské Vinohrady, Šrobárova 1150/50, 100 34 Prague, Czech Republic

**Keywords:** APC, colorectal cancer, EMT, expression profiling, organoids, TACSTD2, WNT/β-catenin signaling

## Abstract

**Simple Summary:**

Colorectal cancer (CRC) is one of the most common cancers worldwide. While the systemic treatment of CRC is based on chemotherapy, subsequent therapeutic options are far less effective. Trophoblast cell surface antigen 2 (TROP2) is highly expressed in many carcinomas, including CRC, where its expression correlates with a poor prognosis. Anti-TROP2-targeted therapy was approved for the treatment of breast and urothelial carcinomas. We aimed to determine whether TROP2 is a suitable target for the treatment of CRC. We demonstrated that TROP2 expression in CRC correlates with lymph node metastasis and poor tumor differentiation. An analysis of mouse tumor models, patient-derived organoids, and tumor cells revealed that TROP2 expression is associated with features related to epithelial–mesenchymal transition and invasiveness. Our results suggest that TROP2 targeting may be a promising approach, especially in the early phase of treatment.

**Abstract:**

Trophoblastic cell surface antigen 2 (TROP2) is a membrane glycoprotein overexpressed in many solid tumors with a poor prognosis, including intestinal neoplasms. In our study, we show that TROP2 is expressed in preneoplastic lesions, and its expression is maintained in most colorectal cancers (CRC). High TROP2 positivity correlated with lymph node metastases and poor tumor differentiation and was a negative prognostic factor. To investigate the role of TROP2 in intestinal tumors, we analyzed two mouse models with conditional disruption of the adenomatous polyposis coli (*Apc*) tumor-suppressor gene, human adenocarcinoma samples, patient-derived organoids, and TROP2-deficient tumor cells. We found that Trop2 is produced early after Apc inactivation and its expression is associated with the transcription of genes involved in epithelial–mesenchymal transition, the regulation of migration, invasiveness, and extracellular matrix remodeling. A functionally similar group of genes was also enriched in TROP2-positive cells from human CRC samples. To decipher the driving mechanism of TROP2 expression, we analyzed its promoter. In human cells, this promoter was activated by β-catenin and additionally by the Yes1-associated transcriptional regulator (YAP). The regulation of TROP2 expression by active YAP was verified by YAP knockdown in CRC cells. Our results suggest a possible link between aberrantly activated Wnt/β-catenin signaling, YAP, and TROP2 expression.

## 1. Introduction

Carcinoma of the colon and rectum (colorectal cancer; CRC) is one of the most common cancers in humans, ranking third in terms of incidence, and is the second leading cause of cancer-related deaths worldwide [1]. In the Czech Republic, despite significant progress in early detection through screening programs, nodal metastases are still diagnosed in about 25% of patients and distant metastases in 20%, with 5-year survival rates of 60% and 15%, respectively [2]. Systemic cancer therapy in patients with CRC metastases is based on chemotherapy consisting of 5-fluorouracil with oxaliplatin or irinotecan in combination with anti-epidermal growth factor receptor (EGFR) or anti-vascular endothelial growth factor (VEGF) monoclonal antibodies. Subsequent lines of therapy are significantly less effective, so there is an urgent need to find new therapeutic targets.

Trophoblast cell surface antigen 2 (TROP2), also known as tumor-associated calcium signal transducer 2 (TACSTD2), is a transmembrane glycoprotein first described as a surface antigen on invasive trophoblast cells [3]. TROP2 is highly conserved, with a 79% identical amino acid sequence between humans and mice and a 49% sequence identity to its closely related homolog epithelial cell adhesion molecule (EPCAM, also known as TASCTD1 or TROP1) [4]. The intronless gene mapped on chromosome 1p32 encodes a 35–46 kDa protein with 323 amino acids (AA), which is composed of a large extracellular domain (AA 1–274), a single transmembrane domain (AA 275–297), and a short cytoplasmic tail (AA 298–323) [5,6]. The extracellular epidermal growth factor (EGF)-like repeat portion starts with a signal peptide (AA 1–26) and contains three structural domains—a cysteine-rich domain, thyroglobulin type-1 domain, and cysteine-poor domain—with four N-glycosylation sites at residues 33, 120, 168, and 208. On the cytoplasmic tail, there is a HIKE domain with a phosphatidylinositol 4,5-bisphosphate (PIP2)-binding motif and a PKC phosphorylation site at serine 303, which upon activation leads to significant conformational changes of the C-terminal tail of TROP2 molecules that form stable dimers [7,8,9].

TROP2 production was found in epithelial cell compartments of the kidney, prostate, pancreas, breast, cervix, uterus, lung, trachea, salivary gland, thymus, and cornea, but was very low or undetectable in the ovary, skeletal muscle, brain, colon, small intestine, liver, thyroid, and spleen [10,11]. Interestingly, inherited mutations in the *TROP2* gene underlie gelatinous drop-like corneal dystrophy (GDLD), a rare autosomal recessive disease characterized by corneal amyloidosis and blindness. The mutations result in the production of the truncated TROP2 protein with decreased expression and altered subcellular localization of tight junction proteins, leading to the decreased barrier function of the corneal epithelium [12].

TROP2 is a putative marker for adult stem/progenitor cells. Most studies have identified TROP2-positive basal cells with stem cell properties in the mouse and human prostate [13,14,15]. Moreover, Trop2 expression defines a specific subpopulation of luminal cells that can enlarge and form large organoids with age in the adult prostate [16]. Not surprisingly, TROP2 marks putative cancer stem cells in prostate cancer [17]. In addition to the prostate, TROP2 has also been identified in stem/progenitor cell compartments such as urinary bladder progenitor cells [18] and airway basal cells [19], and in bipotent hepatic progenitor cells that can form liver organoids containing both hepatocytes and cholangiocytes [20]. Trop2 is also a robust marker of colonic epithelial regeneration during inflammation [21]. In addition, Trop2 has been identified as one of the most important markers for epithelial cells of the developing mouse embryonic gut [22].

The high expression of TROP2 in stem/progenitor cell compartments and in many malignant tumors suggests that TROP2 may regulate important cellular properties common to stem/progenitor cells and/or tumor cells. Several mechanisms of TROP2 signaling have been described. The TROP2 molecule can interact with insulin-like growth factor 1 (IGF1), neuregulin 1, claudin 1/7, and protein kinase C (PKC). The possible signaling pathways triggered by TROP2 are mediated by PIP2 and Ca^++^ ions; these pathways mainly include mitogen-activated protein kinase (MAPK)/extracellular signal-regulated kinase (ERK), AKT, c-Jun N-terminal kinase, and the signal transducer and activator of transcription 3 (STAT3) signaling [11,23,24].

Trop2 activity is post-translationally modulated by several proteolytic cleavages. Trerotola and colleagues showed that the extracellular portion of TROP2 in cancer cells is cleaved by the ADAM10 protease between residues arginine 87 and threonine 88 and that this modification is essential for the metastasis of KM12SM CRC cells [25]. Another enzyme that cleaves TROP2 at the same site is matriptase [26,27]. Another sequential cleavage process, termed regulated intramembrane proteolysis, is mediated by the complexes of TNF-α converting enzyme (between alanine 187 and valine 188) and γ-secretase (between glycine 285 and valine 286). The released intracellular C-terminal TROP2 fragment can accumulate in the nucleus and potentiates β-catenin-dependent signaling. Strikingly, in mouse models, the overexpression of the intracellular fragment in the presence of β-catenin induces prostatic hyperplasia [14]. In gastric cancer, TROP2 promoted epithelial–mesenchymal transition (EMT) by interacting with β-catenin and increasing the expression of mesenchymal markers such as vimentin and fibronectin, while inhibiting E-cadherin expression. The knockdown of TROP2 showed the opposite effect and inhibited metastasis [28]. Recently, it was found that TROP2 binds to CD9 and is subsequently phosphorylated by PKCα, leading to the activation of AKT and ERK, which in turn causes β-actin cytoskeleton remodeling and tumor progression [29].

Due to its high expression in a broad spectrum of human malignancies, TROP2 represents a promising therapeutic target. In 2021, Sacituzumab govitecan, a humanized anti-TROP2 SN-38 monoclonal antibody conjugate, was approved by the U.S. Food and Drug Administration (FDA) for the third-line treatment of patients with metastatic triple-negative breast cancer (TNBC) and urothelial carcinoma [30,31]. Moreover, the role of TROP2 in the mechanisms of resistance to conventional treatments has been observed in many cancers. However, it should be noted that the mechanisms of resistance associated with TROP2 expression are largely unclear. For example, in gastric cancer cells, TROP2 promoted the proliferation and inhibited the chemotherapy-induced apoptosis of cisplatin-treated cells. Interestingly, high levels of TROP2 correlated with an increased expression of Notch1, leading to an upregulation of multidrug resistance protein 1 (MRP1) [32]. In non-small cell lung cancer (NSCLC) cells, treatment with cisplatin induced TROP2 expression accompanied by chemoresistance due to an upregulation of the MAPK signaling pathway [33]. Moreover, EGFR-mutated NSCLC cells with high TROP2 expression developed more rapid resistance to gefitinib therapy through the interaction of TROP2 with insulin-like growth factor 2 receptor (IGF2R), the induction of AKT, and the remodeling of the tumor microenvironment [34]. The latter observation is probably related to the fact that TROP2 expression in cancer cells is a predictor of a response to treatment with AKT inhibitors [35]. Finally, the relationship between TROP2 expression and resistance to oxaliplatin has been described in some CRC cells. It was suggested that the resistance was related to the TROP2-mediated enhancement of EpCAM-induced cell signaling [36].

To investigate the possible role of TROP2 in intestinal tumorigenesis, we examined its expression at both the RNA and protein levels in human preneoplastic lesions, CRC, and in human organoids obtained from healthy colonic epithelia or tumors. In addition, we performed a whole-transcriptome analysis of sorted Trop2-positive neoplastic cells isolated from two models of tumorigenesis in a mouse intestine and from tumor cells obtained from CRC samples. We also disrupted the *TROP2* gene in CRC cells and analyzed the effects of TROP2 deficiency. We found that Trop2 expression is upregulated in early neoplastic lesions arising in the intestines of mice after adenomatous polyposis coli (*Apc*) tumor suppressor gene inactivation; however, this expression is not uniform but is restricted to specific parts of the developing tumors. In CRC patients, a high TROP2 expression was associated with a poor prognosis and correlated with high grading and lymph node involvement. In both mouse and human tumors, TROP2 positivity was associated with EMT. In addition, our analysis of the promoter and expression of *TROP2* in CRC cells suggests the possible coregulation of this gene by aberrant Wnt signaling and Yes1-associated transcriptional regulator (YAP).

## 2. Materials and Methods

### 2.1. Human Adenoma and Adenocarcinoma Samples Used for the Initial Assay of TROP2 Expression

Paired samples of normal and neoplastic colon tissue were obtained from patients who had undergone either polypectomy of colon adenomas or surgical resection of sporadic CRC; sample collection was described previously [37], and patient data are summarized in Appendix A. Human specimen collection methods were performed in accordance with relevant national and EU directives and regulations. The study was approved by the Ethics Committee of the Third Faculty of Medicine of Charles University in Prague (ref. 26/2014 and 01/2020). Informed consent was obtained from all patients who participated in the study. None of the patients underwent radiotherapy or chemotherapy before surgery.

### 2.2. Tissue Microarray (TMA) Analysis

Data on patients (*n* = 292) with surgically resected and histopathologically verified colorectal adenocarcinoma from 2010–2013 with follow-up were obtained from the medical records of the pathology department; the specimen collection was described in the reference [38]. The TMA technique was used to make paraffin blocks using the manual TMA Master instrument (3D Histech, Hungary). Two cylindrical, 2-mm samples were collected from random sites of tumor tissue, and each recipient block contained 20 samples from 10 cases. For immunohistochemistry, 4 µm-thick tissue sections were stained in a Ventana BenchMark ULTRA autostainer (Ventana Medical Systems, Oro Valley, AZ, USA). Monoclonal antibodies against TROP2 (clone EPR20043, Abcam, Cambridge, UK; dilution 1:2000), YAP (clone D8H1X, Cell Signalig, Danvers, MA, USA; 1:200), cytokeratin 7 (CK7; clone OV-TL 12/30, Bio SB, Santa Barbara, CA, USA; 1:500), cytokeratin 20 (CK20; clone Ks20.8, Bio SB; 1:200), special AT-rich sequence binding protein 2 (SATB2; clone EP281, Merck, Kenilworth, NJ, USA; 1:200), and programmed cell death ligand 1 (PD-L1; clone 22C3, pharmDx kit, Agilent, Santa Clara, CA, USA) were used. Positive reactions were visualized using the Ultraview Detection System (Ventana Medical Systems). All immunohistochemical stainings were evaluated by two experienced pathologists (J.H. and R.M.) without knowledge of clinical data and patient history. In each case, the percentage of positively stained TROP2 cells was scored (0 = none, 1–9% = 1, 10–49% = 2, 50–79% = 3, 80–100% = 4), and staining intensity was semi-quantitatively classified into the following four categories: 0 = negative, 1 = weak, 2 = moderate, and 3 = strong positivity. In each case, an overall immunostaining score was calculated as the product of the proportion score and the intensity score (0–3). The total score ranged from 0–12.

### 2.3. Cell Lines

HEK293, SW480, and DLD1 cell lines were purchased from the American Type Culture Collection (cat. nos.: CRL-1573, CCL-228, and CCL-221). HEK293 cells were maintained in Dulbecco’s Modified Eagle’s Medium (DMEM) supplemented with 10% fetal bovine serum (FBS; Thermo Fisher Scientific, Waltham, MA, USA), penicillin, streptomycin, and gentamicin (all antibiotics were purchased from Thermo Fisher Scientific). SW480 and DLD1 cells were maintained in Iscove’s Modified Dulbecco’s Medium (IMDM; Merck) with 10% FBS, penicillin, streptomycin, gentamicin, nonessential amino acids (NEA; Thermo Fisher Scientific), and GlutaMax (Thermo Fisher Scientific).

### 2.4. RNA Purification and Quantitative Reverse Transcription PCR (qRT-PCR)

Total RNA from cell lines and mouse tumors was isolated using TRI Reagent (Merck), and total RNA from sorted cells was isolated using the RNeasy Micro Kit (Qiagen, Germantown, MD, USA); frozen human samples were disrupted in 600 µL of lysis buffer containing green ceramic beads using the MagNA Lyser Instrument (Roche Life Science, Basel, Switzerland). The cDNA synthesis was performed in a 20-µL reaction using random hexamers. RNA was reverse transcribed using MAXIMA Reverse Transcriptase (Thermo Fisher Scientific) according to the manufacturer’s protocol. Quantitative analysis RT-PCR was performed in triplicate using SYBR Green I Master Mix (Roche Life Science) and LightCycler 480 instrument (Roche Life Science). Human samples were analyzed using a combination of gene-specific primers and Universal Probe Library (UPL) hydrolysis probes (Roche Life Science). Threshold cycle (Ct) values for each triplicate were normalized by the geometric mean of the housekeeping genes ubiquitin B (*UBB*) and TATA box binding protein (*TBP*). The resulting values were averaged to obtain ΔCt values for biological replicates. Relative mRNA abundance (ΔCt in healthy tissue–ΔCt in neoplastic tissue) was correlated with histological grade of tumor samples using Spearman (ρ) and Kendall (τ) rank order coefficients. Further details are provided in the corresponding figure legends. A list of primers and UPL probes is provided in Appendix A.

### 2.5. Generation of TROP2 Knockout Cells

To disrupt the human *TROP2* gene, a pair of single-guide RNAs (sgRNA) targeting human *TROP2* were inserted into lentiCRISPRv2 (Addgene, #98293) and corresponding oligonucleotides were inserted into the pARv-RFP reporter vector (Addgene, #60021) [39]. SW480 and DLD1 cells were co-transfected with the lentiCRISPRv2 constructs along with the pARv-RFP reporter vector using Lipofectamine 2000 (Thermo Fisher Scientific) in serum-free OptiMEM medium (Thermo Fischer Scientific). After 48 h, cells expressing red fluorescent protein (RFP) were sorted into 96-well plates and expanded as single-cell clones. Single clones were analyzed by PCR for the presence of deletions in the *TROP2* locus. Amplified genomic DNA fragments were cloned into the pGEM-T Easy vector (Promega, Madison, WI, USA) and sequenced. All sequences are available upon request. Selected clones with both *TROP2* alleles disrupted were further verified by Western blotting. Oligonucleotides and primers are listed in Appendix A.

### 2.6. Generation of Cells with Doxycycline (DOX)-Inducible TROP2 Re-Expression

The sequence of the open reading frame of human *TROP2* was amplified by PCR from cDNA prepared from total RNA isolated from SW480 cells using the following primers: forward 5′-ATGGATCCGGGCAGGTCGGGTAGAGTAT-3′ and reverse 5′-ATCGGTACCCAACAAGCTCGGTTCCTTTCTC-3′. The PCR product was cloned into pFLAG-CMV5a (Merck) using restriction enzymes Bam HI and Kpn I. After sequence verification, the FLAG-tagged *TROP2* coding sequence was cloned into Tet-On lentiviral vector pTripz (Horizon Discovery, Waterbeach, UK) by replacing the tRFP-shRNAmir cassette using Age I and Mlu I restriction enzymes. The resulting DOX-inducible expression vector was transduced into previously prepared TROP2 knockout cells; then, the cells were selected with puromycin (2 μg/mL; Merck). Resistant cells were sorted after TROP2 induction with DOX (final concentration 2 µg/mL; purchased from Merck) by surface labeling with TROP2-specific antibodies (see further).

### 2.7. Cell Viability Test

Cells were seeded in 96-well solid black bottom assay plates (Corning, Corning, NY, USA) at approximately 12.5% confluence in 100 μL IMDM. Six hours after seeding, 10 μL of cell viability reagent alamarBlue HS (Thermo Fisher Scientific) was added to each well. Fluorescence intensity was measured using EnVision^®^ Multilabel Reader (PerkinElmer, Waltham, MA, USA) after 60 min in the culture medium. The measurement was repeated every 24 h for the following four days.

### 2.8. Wound Healing Assay

SW480 and DLD1 *TROP2* gene knockout cells (*n* = 5) and control cells with intact *TROP2* (*n* = 5) were cultured to 100% confluence. Cells were treated with mitomycin C (Merck; final concentration of 10 µg/mL). After 2 h, the cell monolayer was scraped with a 200-µL pipette tip in several straight lines to create a “wound”. Cells were washed with phosphate-buffered saline (PBS), and 2 mL of fresh IMDM was added to each well. For each clone, six positions were photographed immediately and 24 and 48 h after wound formation. The percentage of healed area was quantified using ImageJ software [40].

### 2.9. Mouse Strains Used to Model Intestinal Tumorigenesis

Housing of mice and in vivo experiments were performed in compliance with the European Communities Council Directive of 24, November 1986 (86/609/EEC) and national and institutional guidelines. Animal care and experimental procedures were approved by the Animal Care Committee of the Institute of Molecular Genetics (ref. 58/2017). *Apc^cKO/cKO^* mice [41] were obtained from the Mouse Repository (National Cancer Institute, Frederick, MD, USA); *Villin-CreERT2* mice [42] were kindly provided by Sylvie Robine (Institut Curie, Centre de Recherche, Paris, France); *Apc^+/Min^* (C57BL/6J-ApcMin/J) [43], *Ki67-RFP* (Mki67tm1.1Cle/J) [44], *Lgr5-EGFP-IRES-CreERT2* (B6.129P2-Lgr5tm1(cre/ERT2)Cle/J) [45], and *Rosa26-tdTomato* (B6;129S6-Gt(ROSA)26Sortm14(CAG-tdTomato)Hze/J) [46] mice were purchased from the Jackson Laboratory (Bar Harbor, ME, USA). The animals were kept under specific pathogen-free conditions. To induce Cre-mediated gene recombination, mice were gavaged with 1 mg tamoxifen (Merck). Tamoxifen was dissolved in ethanol (100 mg/mL) and combined (1:9) with mineral oil (Merck) before administration. Mice were sacrificed by cervical dislocation at various time points after administration of a single dose (100 µL) of the tamoxifen/oil solution.

### 2.10. Xenotransplantation

SW480 single-cell clones with *TROP2* gene knockout (*n* = 3) and control cells with intact *TROP2* (*n* = 3) were cultured to 90–100% confluence, harvested, and resuspended in PBS. NSG^TM^-immunodeficient mice (NOD.Cg-PrkdcscidIl2rgtm1Wjl/SzJ; Jackson Laboratory) were injected with 1 million cells in 100 μL PBS into the lumbar region. The mice were sacrificed 28 days after injection, and the tumors were resected and weighed.

### 2.11. Immunofluorescence and Immunohistochemical Staining, Immunoblotting

Intestines were dissected, washed in PBS, fixed overnight in 4% (*v*/*v*) formaldehyde (Merck) in PBS, embedded in paraffin, sectioned, and stained. A detailed protocol of immunohistochemistry and immunofluorescence staining has been described previously [47]. Microscopic images were taken with a Leica DM6000 (wide-field images) or Leica Stellaris confocal platform (both devices were from Leica Microsystems, Wetzlar, Germany). Images were processed using Huygens software (Scientific Volume Imaging, Hilversum, The Netherlands), processed and analyzed using the FiJi package [48]. The immunoblotting procedure was performed as previously described [49]. The primary antibodies that were used include anti-E-cadherin (mouse monoclonal, 610181, BD Transduction Laboratories, Franklin Lakes, NJ, USA), anti-proliferating cell nuclear antigen (PCNA; rabbit polyclonal, ab18197, Abcam, Cambridge, UK), anti-Trop2 (rabbit monoclonal, ab214488, Abcam), anti-Trop2 (goat polyclonal, AF1122, R&D Systems, Minneapolis, MN, USA), anti-vimentin (mouse monoclonal, #5741, Cell Signaling, Danvers, MA, USA), and anti-YAP (4912, Cell Signaling). Secondary antibodies (all from Thermo Fisher Scientific) used include goat anti-mouse Alexa Fluor™ 488 (A-11001), goat anti-rabbit Alexa Fluor™ 488 (A-11034), donkey anti-goat Fluor™ 488 (A11055), donkey anti-rabbit Alexa Fluor™ 594 (A-21207), and goat anti-rabbit biotin-XX (B-2770). Biotin-conjugated secondary antibodies were visualized using the Vectastain Elite ABC-HRP kit-peroxidase (PK-6100, Vector Laboratories, Newark, CA, USA).

### 2.12. Isolation of Intestinal Epithelial Cells

Epithelium was isolated from the proximal jejunum of *Ki67-RFP Apc^cKO/cKO^ Villin-CreERT2* mice 7 days after tamoxifen administration. Microadenomas were isolated from the ileum of *Rosa26-tdTomato-Apc^cKO/cKO^ Lgr5-EGFP-CreERT2* mice 6 weeks after tamoxifen administration. The intestinal tube was cut longitudinally, washed in PBS, and incubated in 5 mM EDTA solution in PBS (pH 8; Merck) for 30 min at 4 °C. After incubation, the solution was shaken vigorously to obtain a suspension of epithelial cells. To isolate crypts from *Apc^cKO/cKO^ Lgr5-EGFP-CreERT2* mice, villi were carefully scraped with the coverslip before incubation in the EDTA/PBS solution. Tumors from 16–18-week-old *Apc^+/Min^* mice were cut directly from the small intestine; healthy control tissue was obtained from *Apc^+/+^* littermates. Tissue cut into small pieces was incubated with dispase (Thermo Fisher Scientific; stock solution 100 mg/mL, diluted 1:300 in serum-free DMEM) on a rotating platform (800× RPM, 5 min, 37 °C). The released cells were transferred to DMEM containing 10% FBS, and the whole procedure was performed three times. Finally, the cell suspension was passed through a 40-µm strainer (Corning). The freshly resected human colon tumor specimens and healthy parts of the colon were mechanically homogenized with scissors. The tissue homogenates were incubated with dispase and processed as mouse tumors. Cell suspension was diluted in DMEM supplemented with 10% FBS and centrifuged at 500× *g* at 4 °C for 5 min to concentrate cells for flow cytometry or organoid cultures.

### 2.13. Organoid Cultures

Epithelial crypts obtained from resected human tissue were embedded in Matrigel (Corning) and cultured as previously described [50]. Advanced DMEM/F12 culture medium (Thermo Fisher Scientific) was supplemented with GlutaMax (Thermo Fisher Scientific), 10 mM HEPES (1M stock, Thermo Fisher Scientific), penicillin/streptomycin (Thermo Fisher Scientific), B27 supplement (Thermo Fisher Scientific), 1.25 mM N-acetylcysteine (Merck), 10 mM nicotinamide (Merck), 50 µg/mL mouse recombinant epidermal growth factor (EGF; Thermo Fisher Scientific), 10 nM [Leu15]-gastrin I (Merck), 500 nM A83-01 (Tocris, Bio-Techne, Minneapolis, MN, USA), 3µM SB202190 (Merck), and 2 µL/mL Primocin^®^ (InvivoGen, Toulouse, France). The medium was also supplemented with mNoggin-Fc [51] and R-spondin 1- (Rspo1) [52] conditioned culture medium (CM) to a final concentration of 10% each CM. Cells producing the indicated secreted proteins were kindly provided by H. Clevers (Hubrecht Laboratory, Utrecht, The Netherlands) and K. Cuo (Stanford University, Stanford, CA, USA). To avoid anoikis, 2.5 mM Y-27632 (Merck) was added to freshly resected crypts for the first 2–3 days and to organoids recovered from liquid nitrogen after thawing. Finally, the culture medium was supplemented with 0.5 nM Wnt Surrogate-Fc Fusion Protein (WntSur; U-Protein Express BV, Utrecht, The Netherlands).

### 2.14. Fluorescence-Activated Cell Sorting (FACS)

Epithelial crypt cells from the ileum of *Ki67-RFP Apc^cKO/cKO^ Villin-CreERT2* mice and microadenomas from *Rosa26-tdTomato Apc^cKO/cKO^ Lgr5-EGFP-CreERT2* mice were stained with Pacific Blue™ (PB) anti-CD45 antibody (#103126, BioLegend, San Diego, CA, USA; dilution 1:200), PB-conjugated anti-CD31 (#102422, BioLegend; 1:200), fluorescein (FITC)-conjugated anti-EPCAM (#11-5791-82, Thermo Fisher Scientific; 1:500), and allophycocyanin (APC)-conjugated anti-TROP2 (FAB1122A, R&D Systems; 1:100) for 20 min at 4 °C; just before sorting, Hoechst 33258 (Merck) was added to the cell suspension. Cells were sorted by forward scatter (FSC), side scatter (SSC), and negative staining for Hoechst and PB. EPCAM^+^ (epithelial) cells were further sorted for Trop2 expression to obtain EPCAM^+^Trop2^+^ and EPCAM^+^Trop2^−^ cells. The red fluorescence of RFP/tdTomato was used to distinguish proliferating and tumor cells. Cell sorting was performed using the Influx Cell Sorter (BD Biosciences, San Jose, CA, USA). Human colon cells were stained with PB-conjugated anti-CD45 (#304022, BioLegend; 1:400), PB-conjugated anti-CD31 (#303114, BioLegend; 1:400), PE-conjugated anti-EPCAM (FAB9601P, R&D Systems; 1:500), and APC-conjugated anti-TROP2 (FAB650A, R&D Systems; 1:100) and with Hoechst 33258 20 min at 4 °C. Cells were sorted by FSC, SSC, and negative staining for Hoechst and PB; EPCAM^+^ cells were sorted according to TROP2 expression.

### 2.15. Bulk RNA Sequencing (RNA-seq)

Sequencing libraries were prepared from total RNA using the Smarter Stranded Total RNA-seq Kit v2 Pico Input Mammalian (Takara, Japan), followed by size distribution analysis in the Agilent 2100 Bioanalyzer using the High Sensitivity DNA Kit (Agilent). Libraries were sequenced in the Illumina NextSeq 500 instrument (Illumina, San Diego, CA, USA) using a 76 bp single-end configuration. Read quality was assessed using FastQC (Babraham Bioinformatics, Babraham Institute, Cambridge, UK). The nf-core/rnaseq bioinformatics pipeline [53] was used for subsequent read processing. Individual steps included removal of sequencing adapters and low-quality reads with Trim Galore! (Babraham Bioinformatics), mapping to the reference GRCm38 genome (Ensembl 2019) using HISAT2 [54], and quantifying gene-level expression with featureCounts [55] (mouse cell analysis) or quantification of expression with Salmon [56] using GRCh38 as a reference (human cell analysis). Expression per gene served as input for differential expression analysis using the DESeq2 R Bioconductor package [57] performed separately for each experiment. Genes not expressed in at least two samples were screened out before analysis. We created an experimental model in which the sample group was assumed the main effect while considering the identity of the samples. Genes that had a minimum absolute log_2_ fold change of 1 (|log_2_| FC ≥ 1) and statistical significance (adjusted *p*-value < 0.05) between the compared sample groups were considered differentially expressed.

### 2.16. Raw Expression Data Repository

Minimum Information About a Microarray Experiment (MIAME)-compliant data were deposited to the ArrayExpress database under accession numbers: E-MTAB-11377 (analysis of *Ki67-RFP Apc^cKO/cKO^ Villin-CreERT2* mice), E-MTAB-11382 (*Rosa26-tdTomato Apc^cKO/cKO^ Lgr5-EGFP-IRES-CreERT2* mice), and E-MTAB-11466 (human specimens).

### 2.17. Luciferase Reporter Assays

A putative promoter region of the human *TROP2* gene consisting of a 3 kbp sequence upstream of the transcription start site was analyzed in silico with the CiiiDER web tool [58] for the presence of T-cell factor (TCF)/lymphoid enhancer factor (LEF) and TEA domain (TEAD) transcription factor binding sites using previously published position frequency matrices [59]. Proposed binding sites with low match to consensus binding sequences (parameters: Core Match Score below 1 and Matrix Match Score below 0.93 and 0.858 for TCF/LEF and TEAD, respectively) were subsequently filtered out. A 2765-bp sequence from the upstream regulatory region of *TROP2*, which includes three putative TCF/LEF and five TEAD binding sites, was amplified from HEK293 genomic DNA with the primers forward 5′-GTTTACATAATAAACTCATTGTGGTCTTTGT-3′ and reverse 5′ TATAGTTTACAGTTCACAAACATTATCTCATC-3′ and was then cloned into the pGEM-T Easy vector (Promega). After sequence verification, the fragment was subcloned into the pGL4.26 luciferase reporter (Promega) using cloning sites Xho I and Bgl II. HEK293, DLD1, and SW480 cells were transfected with 0.1 ug of reporter, 0.05 ug of pRL-TK control vector (Promega), and expression/empty vector or 0.55 ul of 20μM corresponding siRNA in 24-well plates using Lipofectamine 2000 (Thermo Fisher Scientific) according to the manufacturer’s protocol. Cells were incubated for 48 h after transfection and lysed with Passive Lysis Buffer (Promega). Dual reporter assay was performed using the Luciferase Assay System (Promega) and the Renilla Luciferase Assay System (Promega) in the Glomax 20/20 Luminometer (Promega). Reporter vectors included: pGL4.26 with the *TROP2* promoter; Wnt reporter 8xSuperTopFlash, a gift from Randall Moon [60]; and YAP/TAZ reporter 8xGTIIC (Addgene #34615), a gift from Stefano Piccolo [61]. Expression vectors included: S127A YAP [62] (Addgene #27370)—a kind gift from Kunliang Guan—and S45A β-catenin [63]. For RNAi, cells were transfected with Lipofectamine RNAMax (Thermo Fisher Scientific) with 10 nM siRNAs: siCTNNB (#M-003482-00-0005, Horizon Discovery, Waterbeach, UK), siYAP (#M-012200-00-0005, Horizon Discovery), and non-targeting siRNA control (#D001206-13-20, Horizon Discovery). Experiments were performed in triplicate; luciferase values were normalized to the corresponding Renilla luciferase values and plotted as relative luciferase units (RLU) in Prism (GraphPad Sofware, La Jolla, CA, USA).

### 2.18. Statistical Analysis

Statistical analysis was performed using R, version 4.1.2. Exploratory data analysis was performed for all parameters. Data are presented as mean ± standard deviation (SD; normally distributed data), median and interquartile ranges (non-normally distributed data), and counts with frequencies for categorical data. Correlations between TROP2 scoring groups and clinicopathological variables after binarization were examined using univariate logistic regression. Survival probabilities for cancer-specific survival were determined using the Kaplan–Meier method and the log-rank test, and confidence intervals were calculated using the log–log method. Multivariate Cox regression was used to test the prognostic significance of TROP2 expression and other parameters. Results are presented as odds ratio/hazard ratio with 95% confidence intervals. A *p*-value of less than 0.05 was considered statistically significant.

## 3. Results

### 3.1. TROP2 Expression Is Upregulated during Neoplastic Transformation of Human Colonic Epithelium

To investigate the expression of TROP2 during tumorigenesis in the colonic epithelium, we collected and analyzed 70 paired endoscopic biopsy samples at different stages of neoplastic transformation. A quantitative RT-PCR analysis revealed that the TROP2 gene expression was significantly increased in all stages of the tumor-formation process compared with the healthy controls, i.e., in hyperplastic polyps, adenomas with an increasing degree of dysplasia, and CRCs (Figure 1A). Subsequently, the localization of the TROP2 protein was detected via immunohistochemistry. In contrast to the normal colonic epithelium, which did not show TROP2 staining, the preneoplastic lesions showed a clear albeit “patchy” TROP2 expression (Figure 1B). In CRC tissue, TROP2 showed a heterogeneous staining intensity ranging from negative to strong expression in almost all malignant cells, and the corresponding metastases exhibited the staining pattern of the primary tumor (Figure 1C).

### 3.2. TROP2 Is an Independent Negative Prognostic Factor in CRC Patients

Few prognostic molecular markers have been validated in patients with CRC [64], and only a limited number of available options are used in clinical practice for CRC targeting. Therefore, there is a constant need to search for new predictive and prognostic CRC molecular markers. TROP2 is considered a promising therapeutic target in several other human malignancies [65]. Therefore, we used TMA to investigate the prognostic value of TROP2 in 292 retrospective CRC cases. The TROP2 expression was assessed by the total immunostaining score (TIS), which was calculated as the product of the proportion score and the staining intensity score with the total value ranging from 0 to 12. A photomicrograph of TMAs with four different scoring categories is shown in Appendix A; the distribution of TISs within each category is shown in Appendix A. To better characterize the cases of CRC, we determined four immunohistochemical markers in addition to the standard clinicopathological parameters: PD-L1, CK7, CK20, and SATB2. Programmed death ligand 1 is considered a predictive marker for immune checkpoint inhibitors already approved for the treatment of microsatellite instability-high (MSI-H) CRC. Intermediate filament component CK7 was found to be a negative prognostic factor in a subset of CRC [38]. Cytokeratin 20 is a routinely used diagnostic marker for CRC, and the combined immunostaining of CK20 and SATB2 has been shown to distinguish the majority of CRC [66]. In addition, CK20 and SATB2 are considered markers of colorectal differentiation, and the loss of SATB2 expression is associated with an aggressive phenotype of CRC and poor prognosis [67].

For survival analysis, we used the Kaplan–Meier method and the log-rank test. The patient cohort was first divided into three subgroups according to their TISs: a score of 0–4 (low expression) in 179 cases, a score of 6–8 (medium expression) in 44 cases, and a score of 9–12 (high expression) in 69 cases. Considering the “flattening” of the survival curves at the 5-year follow-up for the low and medium expression scores (Appendix A), we combined these scoring categories into two groups: low/medium expression (score 0–8) in 223 (76.4%) cases and high expression (score 9–12) in 69 (23.6%) cases (Appendix A). The patient data and correlation analysis of the TROP2 expression with the clinicopathological characteristics are summarized in Appendix A, and the statistically significant variables are also shown in bar charts (Appendix A). Using a univariate logistic regression, a high TROP2 expression was significantly correlated with a poor histological grade, mucinous and signet ring cell morphology, lymph node metastasis, an advanced stage III + IV, localization in the right colon, and positive staining for PD-L1 and CK7. An inverse correlation with SATB2 staining was also observed. Patients with a high score had a significantly shorter cancer-specific survival (CSS) than the low/medium score group, as shown by the Kaplan–Meier survival curves (Figure 2A). The restricted median survival time (RMST), i.e., the median survival from surgery to CRC-related death, was 5.93 years [95% confidence interval (CI); 4.92–6.94] in patients with high TROP2 expression compared to 7.68 years (95% CI; 7.20–8.15) in the low/medium score group (*p* = 0.0012); the RMST values were obtained for 10 years of follow-up and are shown in Figure 2B. The same results were observed in the survival analysis based on the TROP2 proportion score with an optimal cutoff value of 25% positively stained cancer cells (*p* = 0.00016), and the corresponding survival curve is shown in Figure 2C. A multivariate Cox regression identified high TROP2 expression together with lymph node metastases and an age older than 75 years as an independent negative prognostic factor for poor CSS, whereas CK7 reached borderline significance (Figure 2D).

### 3.3. TROP2 Expression in CRC Cells Is Associated with Gene Signatures Related to EMT and Cell–Extracellular Matrix Interaction

To determine the gene expression associated with TROP2 surface expression on CRC cells, we performed gene expression profiling of cells isolated from human CRC samples. Live epithelial cells (labeling: Hoechst33358/CD31/CD45^−^ EPCAM^+^) were sorted based on TROP2 protein abundance from single-cell suspensions obtained from freshly resected human CRC samples (Figure 3A). Cells from four CRCs were subsequently analyzed by bulk RNA-seq. A principal component analysis (PCA), based on analysis of the 500 most differentially expressed genes, revealed considerable heterogeneity between tumors. This was reflected by the greater similarity of expression profiles between TROP2^high^ and TROP2^low^ cells isolated from the same tumor than when comparing the expression profiles of TROP2^high^ (or TROP2^low^) cell populations derived from different tumors. Nevertheless, in all the samples, TROP2 expression levels were elevated in the TROP2^high^ cells compared with TROP2^low^ cells, confirming the sorting strategy (Appendix A). We then performed a gene set enrichment analysis (GSEA) of 72 genes significantly (adjusted *p*-value < 0.05) upregulated in TROP2-producing cells using the Enrichr online tool [68]. We used either the Molecular Signatures Database (MSigDB) Hallmark 2020 [69] or Gene Ontology (GO) Biological Processes 2021 [70,71] gene set collections.

The highest-scoring signaling pathways in the TROP2^high^ cells according to the MsigDB Hallmark 2020 gene sets included EMT, KRAS signaling, TNF-α signaling via NF-κB, and intereleukin (IL) 2/STAT5 signaling (Figure 3B). The corresponding genes involved in EMT and extracellular matrix reorganization that were upregulated in the TROP2-positive cells were laminin subunit alpha 3 and gamma 2 (*LAMA3* and *LAMC2*), filamin A (*FLNA*), dystonin (*DST*), C-X-C motif chemokine ligand 1 (*CXCL1*), vascular endothelial growth factor A (*VEGFA*), osteopontin (*SPP1*), plasminogen activator-like urokinase receptor (*PLAUR*), disintegrin and metalloproteinase 19 (*ADAM19*), and matrilysin [(also known as matrix metalloproteinase 7 (*MMP7*)] (Appendix A).

### 3.4. Trop2 Expression Marks a Subset of Tumor Cells in Two Mouse Models of Intestinal Tumorigenesis

To analyze the Trop2 expression in the mouse tumor tissues, we employed *Apc^Min/+^* mice. In the genome of these mice (originally designated as mice with multiple intestinal neoplasia (Min)), one allele of tumor suppressor *Apc* is inactivated by mutation. The incidental inactivation of the second functional allele of this gene in the intestinal epithelium results in the nonphysiological activation of the Wnt signaling pathway, which is associated with the development of intestinal tumors [43]. The mouse strain is commonly used to study intestinal tumorigenesis associated with the hyperactivation of the Wnt pathway [72]. We observed a clear, albeit scattered, Trop2 protein positivity in the tumors in the small and large intestine. In contrast, the staining in healthy tissue was negligible (Appendix A). Quantitative RT-PCR profiling confirmed high *Trop2* mRNA levels in the tumors derived from different parts of the small intestine and colon. In addition, a slight increase in *Trop2* expression was observed in the middle and distal part of the healthy small intestine compared with the proximal small intestine segment and colon (Appendix A).

To monitor Trop2 expression in early neoplastic intestinal lesions, we used the conditional (cKO) allele of the *Apc* gene [41]. We first knocked out the gene in all the intestinal epithelial cells by crossing *Apc^cKO/cKO^* mice with the *Villin-CreERT2* mouse strain [42]. Alternatively, *Apc* was inactivated in the intestinal epithelial stem cells (IESCs) by using the *Lgr5-CreERT2-EGFP* strain [45]. The histological analysis of *Apc^cKO/cKO^ Villin-CreERT2* mice performed seven days after tamoxifen administration, i.e., Apc inactivation, revealed hyperplastic epithelia with Trop2 positivity mainly on the villi. In the (micro)adenomas that formed in the *Apc^cKO/cKO^ Lgr5-CreERT2-EGFP* mice three weeks after *Apc* gene inactivation, we also observed that some areas of the tumors produced Trop2 (Figure 4A). To investigate the properties of the Trop2-producing tumor cells, we used two (additional) mouse strains. First, we used *Ki67-RFP* mice that produce a Ki67-RFP fusion protein from the endogenous *Ki67* locus [44]. After their crossing with *Apc^cKO/cKO^ Villin-CreERT2* mice and an administration of tamoxifen, early neoplastic lesions developed in the intestines of these animals. As most tumor cells divide, the cells of these lesions produce the proliferation marker Ki67 and are simultaneously labeled by the red fluorescence of RFP protein (Figure 4B, left part of the diagram). In the case of the second strain, we used the so-called reporter allele *R26-tdTomato*, which has an integrated gene encoding a tandem dimer (td) of the red fluorescent protein Tomato in the *Rosa26* locus. Although this locus is transcriptionally active in all mouse cells, the tdTomato protein is not translated because of the insertion of a transcription blocker upstream of the translation-initiation codon of the *tdTomato* gene, which prevents the transcription of *tdTomato* mRNA. Transcription (and production of the tdTomato protein) then occurs only in the cells in which the blocker has been “excised” by Cre recombinase [46]. Cre-mediated recombination in a second mouse strain, *R26-tdTomato Apc^cKO/cKO^ Lgr5-EGFP-CreERT2*, resulted in the knockout of the *Apc* gene in IESCs, accompanied by the production of the red fluorescent protein tdTomato in growing adenomas (Figure 4B, right part of the diagram). Stereoscopic microscopy images of lesions that developed in the midportion of the small intestine of the *Ki67-RFP Apc^cKO/cKO^ Villin-CreERT2* and *R26-tdTomato Apc^cKO/cKO^ Lgr5-EGFP-CreERT2* mice 7 days and 6 weeks after tamoxifen administration, respectively, are shown in Figure 4C (left panels). Interestingly, the immunofluorescence staining of the marker for proliferating cells, PCNA, showed that only a minority of proliferating tumor cells produced Trop2 (Figure 4C, right panels). First, we isolated RFP-positive proliferating cells from the hyperproliferative epithelium of *Ki67-RFP Apc^cKO/cKO^ VillinCreERT2* mice 7 days after Cre-mediated recombination and sorted them according to their Trop2 expression. To verify a correct cell sorting, we also isolated a fraction of nonproliferative, i.e., RFP-negative, cells that contained mainly differentiated Apc wild-type (WT) epithelial cells. Similarly, we used *R26-tdTomato Apc^cKO/cKO^ Lgr5-EGFP-CreERT2* mice 6 weeks after Apc inactivation to isolate tdTomato^+^Trop2^+^ and tdTomato^+^Trop2^−^ tumor cells and additionally a tdTomato-negative cell fraction that presumably contained differentiated Apc WT small intestinal epithelial cells (Figure 4D). The expression profiling of the isolated cell populations was then performed.

A comparison of the expression profiles of Trop2^+^Ki67^−^RFP^+^ and Trop^−^Ki67^−^RFP^+^ cells with cells that did not produce the Ki67-RFP fusion protein revealed that the genes that are highly overproduced in “red” cells have functional relationships with various stages of the cell cycle, such as DNA replication and mitosis. Moreover, tdTomato-positive cells in the adenomas of *R26-tdTomato Apc^cKO/cKO^ Lgr5-EGFP-CreERT2* showed a significant upregulation of Wnt target genes compared with tdTomato-negative cells [73,74,75,76]—also see the webpage https://web.stanford.edu/group/nusselab/cgi-bin/wnt/target_genes (accessed on 1 February 2022). In addition, the cells labeled with Ki67-RFP showed a significantly decreased expression of IESC marker olfactomedin 4 (*Olfm4*). This indicated that the Ki67-RFP^+^ cell population predominantly contained tumor cells and was not “contaminated” by healthy epithelial crypt cells (Appendix A). The subsequent analysis showed that there was a significant difference (the significance criterion: (|log_2_ FC| ≥ 1; adjusted *p*-value < 0.05) in the expression of a total of 2944 genes between the Trop^+^ and Trop^−^ tumor cells; 240 genes were identified in both experiments (Figure 5A, Appendix A). We then performed a GSEA of the genes whose expression was elevated in the Trop2-producing cells using the online tool Enrichr. As with the human samples, we used either the MSigDB Hallmark 2020 gene set or the GO Biological Processes 2021 gene set. The analysis based on the MSigDB Hallmark 2020 database showed that mainly genes related to EMT were enriched in the Trop^+^ tumor cells obtained from both mouse models (Figure 5B). Surprisingly, a dataset related to the coagulation process scored second in the Trop-positive cells obtained from adenomas and third in the Trop^+^ cells present in hyperplastic lesions. On closer analysis, we found that this was due to the increased expression of several extracellular and intracellular proteases involved in the coagulation cascade, among others (Appendix A).

In addition, genes involved in TNF-α signaling via NF-κB were enriched in Trop2-positive cells in both tumor models. Additional datasets included hypoxia and active KRAS and IL2/6-STAT/5/3 signaling; furthermore, apoptosis and myogenesis were also indicated (Figure 5B). In the GO Biological Processes 2021 database, the genes falling into the categories ‘Extracellular matrix/structure organization’ and ‘Regulation of cell proliferation’ were the most significantly enriched in Trop2^+^ cells obtained from hyperplastic lesions or adenomas, respectively. Moreover, the genes (over)expressed in the Trop2^+^ cells represented the processes related to cell migration and the positive regulation of cell motility (Trop2^+^ cells from hyperplastic lesions) or the positive regulation of cell proliferation, cell motility, the ameboidal type of cell migrations, and cell to cell adhesion (Trop2^+^ cells from adenomas; Figure 5C). Vimentin is an intermediate filament involved in cell migration, motility, and adhesion. In solid cancers, vimentin controls EMT [77]. Given the results of our bulk RNA-seq experiment showing that Trop2-positive cells in tumors exhibit an EMT signature, we colocalized Trop2 and vimentin in transformed intestinal epithelial cells (Figure 5D). We identified annexin A1 (*ANXA1*) and A-kinase anchoring protein 12 (*AKAP12*) as genes enriched in both mouse and human TROP2-positive/high cells (Appendix A).

### 3.5. TROP2 Expression Level in Human Colon Organoids Derived from Healthy Epithelium Depends on Wnt Pathway Activity

We then examined the expression of TROP2 in organoids derived from human CRCs; the healthy tissue adjacent to the tumor was used to establish cultures of “healthy” organoids. Initially, the organoids were cultured in a standard ENR medium containing EGF, noggin, and RSPO1 without the addition of a Wnt ligand. This medium does not support the long-term growth of healthy organoids, but the established organoids can survive in a culture for about two weeks. However, the ENR medium is suitable for the growth of tumor organoids for which the stimulation of division and growth by the exogenous Wnt ligand is not required [78]. To increase the stimulation of the Wnt pathway, the next-generation surrogate Wnt ligand (WntSur) was added to the ENR medium at increasing concentrations. Organoid lines derived from tumor and healthy tissues of three patients were analyzed. In the healthy organoids, the TROP2 expression levels were quite similar and increased with the stimulation of the Wnt pathway. The same trend was observed for the Wnt target genes and vimentin. In contrast, the increasing stimulation of the Wnt pathway resulted in the decreased expression of differentiated cell markers, alkaline phosphatase (*ALPI*), and sucrase isomaltase (*SIM*), as well as of epithelial cell marker E-cadherin. In tumor organoids, the situation was different. The level of *TROP2* gene expression in the organoids obtained from tumor tissue differed markedly from patient to patient, and an enhanced stimulation of the Wnt pathway had no effect on this expression, as confirmed by the immunofluorescence detection of TROP2 in the tumor organoids cultured in an ENR medium without the addition and with the addition of the WntSur ligand. The expression of the other genes tested was also unaffected by an enhanced activation of the Wnt pathway (Figure 6A). Previous work suggests that Trop2 is expressed in the intestinal epithelium under certain conditions, particularly during embryonic gut development and in the recovery phase after tissue damage [21,22]. It is known that under these conditions, cell growth and tissue regeneration are driven by the YAP-tafazzin (TAZ)-driven transcriptional program [22,79,80,81]. The transcriptional activity of YAP and TAZ is controlled by the Hippo signaling pathway. The Hippo signaling pathway was originally discovered in Drosophila melanogaster as a system for regulating organ size; however, its core components are evolutionarily conserved [82,83]. When the Hippo signaling pathway is inactive, the YAP and TAZ mediators are translocated to the nucleus, where they interact with members of the TEAD transcription factor family and activate the transcription of various pro-proliferative and anti-apoptotic genes [84]. We performed an analysis of the expression of the (putative) target genes of YAP/TAZ signaling. In addition, we analyzed the localization of YAP in the organoid cells. The results of the qRT-PCR analysis confirmed significant differences between the expression profiles of the tumor organoids.

In the healthy organoids, six of the eight potential target genes of the YAP/TAZ pathway tested responded in a significant manner by an increase in expression upon increasing concentrations of WntSur ligand. The *ANXA1* gene showed no response, and for the *CTSE* gene, we observed a decrease in expression. Interestingly, the immunofluorescence detection of the YAP protein in organoids obtained from healthy colon epithelium and grown at a high WntSur concentration showed the nuclear positivity of the YAP protein in most cells (Figure 6B).

### 3.6. The TROP2 Promoter Is Activated by Transcriptional Regulator YAP

Next, we tested the possibility of the direct regulation of the *TROP2* promoter by YAP/TAZ. At the same time, we considered that TROP2 expression is (co)regulated by Wnt/β-catenin signaling. According to the canonical mode of action, YAP/TAZ activates the transcription of target genes in complex with a transcription factor from the TEAD protein family [62,85]. We performed an in silico search for binding motifs for both TEAD and TCF/LEF transcription factors in the 2.7-kbp region upstream of the TROP2 gene transcription start site using the freeware tool CiiiDER [58]. We localized five TEAD binding sites and three TCF/LEF binding sites (Figure 7A). We also cloned the promoter region containing all the detected binding sites into a luciferase reporter vector and performed a series of luciferase assays. In the HEK293 cells, we tested the response of the *TROP2* promoter region to the stimulation of the canonical Wnt pathway by Wnt3a-conditioned media and the inhibition of GSK3 kinase by BIO. We also measured the promoter response in HEK293 cells to the transient expression of transcriptionally active nonphosphorylated β-catenin S45A or YAP variant S127A. Luciferase assays using the synthetic SuperTopflash (STF) [60] reporter for Wnt signaling and the YAP/TAZ-responsive reporter 8xGTIIC (abbreviation GT2) [61] were employed to monitor the functionality of the stimuli used.

No response of the *TROP2* promoter to an exogenous stimulation by Wnt3a CM was detected. The hyperactivation of the Wnt pathway by the GSK3 inhibitor BIO or by the expression of the stable oncogenic β-catenin variant S45 only moderately increased the *TROP2* promoter activity (Figure 7B). However, in the cells expressing S127A YAP, a significant increase in luciferase activity (driven by the *TROP2* promoter) was observed (Figure 7C). In addition, we used APC-deficient DLD1 and SW480 cells to perform an siRNA-mediated knockdown of β-catenin (encoded by the *CTNNB* gene), YAP, and TAZ, followed by a luciferase assay. Interestingly, the knockdown of *CTNNB* mRNA increased the luciferase expression of the *TROP2* promoter, whereas YAP or simultaneous knockdown of YAP and TAZ decreased the *TROP2* promoter activity in DLD1 and more significantly in the SW480 cells (Figure 7D,E). Moreover, qRT-PCR analysis clearly showed that *TROP2* expression was reduced after the knockdown of *YAP* or *TAZ* or both mRNAs simultaneously; the functionality of the experiment was confirmed by the reduced expression of the target gene of YAP/TAZ signaling, *CYR61*. Similar to the reporter experiments, the knockdown of β-catenin increased *TROP2* expression. The expression of the control gene for Wnt/β-catenin signaling, *AXIN2*, was significantly reduced after the knockdown of the mRNA-encoding β-catenin (Figure 7F,G). This would indicate a possible interference between YAP and β-catenin in CRC cells.

Cells with an active transcription of YAP/TAZ target genes are characterized by YAP speckles localized in the nucleus, which presumably represent the active fraction of the YAP protein [86]. Interestingly, using immunofluorescence staining, we observed sparse spots of the nuclear staining of YAP in WT intestinal crypts and we did not see the YAP signal in the differentiated WT cells on the villi. However, there were clearly visible YAP speckles in the nuclei of the cells in the Trop2-positive areas of the epithelium. This was most pronounced in the transformed cells that formed clusters of cells with high Trop2 membrane positivity in adenomas and at the tips of hyperplastic villi (Figure 8). We next investigated whether the colocalization of TROP2 and YAP also occurs in human tumors by analyzing the selected CRC cases (Figure 9). In contrast to TROP2 immunostaining, the cytoplasmic and scattered nuclear expression of YAP was detected in the majority of the samples examined. In some cases, nuclear YAP positivity colocalized with a strong membranous TROP2 staining (Figure 9A,B). In other cases, the overlap was observed only in limited areas of the tumor, including neoplastic cells forming the invasive margin in lymph node metastases (Figure 9C,D). However, we also observed a heterogeneous nuclear YAP positivity that was not followed by detectable TROP2 expression (Figure 9E,F).

### 3.7. TROP2 Deficiency Increases the Migratory Ability of CRC Cells 

To investigate the cellular processes associated with TROP2, we disrupted the *TROP2* gene in human DLD1 and SW480 CRC cells using the CRISPR/Cas9 system. The inactivation of the *TROP2* locus was verified by sequencing genomic DNA and immunoblotting the TROP2 protein (Appendix A). The TROP2-deficient (KO) cells were viable and showed no difference in their proliferation rate compared with the TROP2-proficient control cells (Appendix A, left). To examine the proliferation in vivo, we subcutaneously injected immunodeficient NSG mice with SW480 TROP2 KO and TROP2 WT cells. We observed a trend toward smaller tumors in the TROP2 KO cell clones, although it was not statistically significant (Appendix A, right). Next, we performed a “wound healing assay” to determine whether TROP2 had an effect on cell migration. The assay showed an increased migration of SW480 KO cells compared to WT cells, but no difference in migration was observed between the DLD1 KO and WT cells (Appendix A). To validate the results of the assay (obtained in SW480 cells), we generated SW480 iTROP2 cells with the doxycycline (DOX)-inducible re-expression of TROP2 (Appendix A). Consistent with previous results, iTROP2 cells cultured without DOX showed an increased migration compared with cells treated with DOX, i.e., the re-expression of TROP2 reduced cell migration (Appendix A).

## 4. Discussion

Transmembrane glycoprotein TROP2, identified as a surface marker for invasive trophoblast cells, is overproduced in many tumor types. TROP2 positivity in tumors is associated with the increased proliferation, migration, and invasiveness of tumor cells, which is associated with a poorer prognosis [87]. In this study, we demonstrated that *TROP2* gene expression increases in the preneoplastic stages of adenomas with low-grade dysplasia and that this expression was observed in 88% of all CRC cases. Trerotola and colleagues have previously described an increased expression of TROP2 in dysplastic intestinal epithelium compared to normal intestinal mucosa, in which TROP2 is undetectable by immunohistochemistry [88]. In our study, we confirmed this increased expression of TROP2 in adenomas with low- and high-grade dysplasia and additionally observed an increased expression of TROP2 in the epithelia of hyperplastic polyps. When comparing the TROP2 expression profile in the tumor samples from the patients, we observed an apparent discrepancy between the relatively small increase in *TROP2* mRNA expression in LGD, HGD, and CRC tumors and the relatively strong positivity of TROP2 protein/antigen staining in the same tumor type (Figure 1). We suggest that this discrepancy is due to the relative stability of the TROP2 protein observed in some cell lines derived from CRC [89]. It should also be noted that the antibody used for its detection recognizes the intracellular portion of TROP2, including the C-terminal fragment. Therefore, the staining of TROP2 in tumor samples is not affected by the cleavage of the extracellular domain, which is frequently detected in tumor cells [25]. Further data demonstrating that TROP2 is upregulated in the early stages of tumorigenesis were provided by Riera and colleagues in a study using mouse models of gastric tumorigenesis and human biopsy samples, which showed that TROP2 is a potential biomarker for the transition from gastric metaplasia to dysplasia and adenocarcinoma of the stomach [90]. The role of Trop2 in the earliest stages of tumorigenesis has also been described in the prostate, where it triggers hyperplasia via Wnt/β-catenin signaling [14]. 

A high *TROP2* gene expression, as determined by a combined immunohistochemical score of 9–12 points, was detected in approximately one quarter of the CRC samples analyzed and represents an independent negative prognostic factor. This is consistent with previous data on the prognostic role of TROP2 in CRC patients. Ohmachi and colleagues demonstrated that *TROP2* mRNA expression was significantly increased compared with healthy intestinal tissue; in more than one-third of cases (74 CRC cases were analyzed in total), a high expression of TROP2 also correlated with liver metastases and poor prognosis [91]. Using the immunohistochemical detection of a tissue microarray of 620 CRC samples, Fang and colleagues showed that the positivity of TROP2 correlated with liver metastases, tumor recurrence, and poor prognoses [92]. Finally, in a cohort of 80 colorectal tumors, Guerra and colleagues showed that TROP2 protein positivity was correlated with lymph node metastasis. In a second cohort of 53 CRC patients with up to 400 months of follow-up, the same authors showed that a high expression of TROP2 was associated with a poor prognosis for overall survival [93]. In our cohort of 292 patients with 10-year survivals, we found a significant association of high TROP2 protein production not only with metastatic lymph node involvement, but also with poor differentiation (grade 3) and mucinous and signet ring tumor morphologies. This was supported by an inverse correlation with differentiation markers CK20 and SATB2. In addition, we observed a positive correlation between a high TROP2 expression and CK7, which tends to be upregulated in cancers with increased migration, invasion, and metastatic ability [94] and with PD-L1 expression (≥1%) on tumor cells, which is associated with induced immune tolerance and poorer prognoses [95,96]. In our previous study, using a DNA microarray-based expression screen, we identified *Trop2* as a gene whose expression is significantly increased in the small intestinal epithelia of mice after the knockout of the tumor suppressor gene *Apc* [47]. To determine the site of expression and the molecular mechanisms that trigger an increased TROP2 expression in the early stages of intestinal neoplasia, in the present study, we analyzed murine intestinal tumors induced by the conditional inactivation of *Apc*. The gene was inactivated either throughout the epithelium or in IESCs. Interestingly, similar to the expression of the *Msx1* gene, which is also activated in intestinal tumors after Apc loss, the expression of the *Trop2* gene is not uniform in tumors, but its production is restricted to specific tumor cells [47]. Depending on the Cre driver used, Trop2-positive cells are found in villi (Cre driver: Villin-CreERT2) or in specific regions of microadenomas (Cre driver: Lgr5-EGFP-IRES-CreERT2). This suggests that even very early neoplastic lesions contain heterogeneous tumor cells.

We used RNA sequencing to determine the expression profile of TROP2-positive cells. This profile includes a group of genes mainly involved in EMT, the regulation of migration, invasiveness, and the reorganization of the extracellular matrix. These genes encode the extracellular matrix proteins *LAMA3* and *LAMC2*, which have been detected in invasive colon cancer cells and are regulated by transforming growth factor beta 1 and the hepatocyte growth factor produced in the tumor microenvironment [97]. Another EMT marker/promoter, *Spp1*, is highly expressed in liver metastases [98] and promotes tumor cell migration and invasion [99]; *PLAUR* is responsible for the binding and conversion of plasminogen to active plasmin and the subsequent degradation of the extracellular matrix, which in turn promotes invasion and metastasis [100]. Finally, matrilysin is a matrix metallopeptidase that can degrade extracellular matrix components such as laminins, fibronectin, collagens, and proteoglycans and promote tumor cell invasion and metastatic spread [101]. 

To confirm these results, we also determined the expression profile of TROP2-high cells isolated from human tumor samples. A group of genes involved in EMT, the cell–substrate interaction, and extracellular matrix remodeling were also more highly expressed in the cells with a high TROP2 production than in tumor cells with a low TROP2 production. Two genes, *ANXA1* and *AKAP12*, were enriched in both mouse and human TROP2-producing tumor cells. Annexin A1 is a calcium-dependent phospholipid-binding protein that plays a variety of roles in the immune system and cancer [102]. In several cancer types, ANXA1 is involved in processes regulating increased invasiveness and EMT [103,104]. In CRC, ANXA1 is associated with invasion and lymph node metastasis [105]. AKAP12 is a scaffold protein that plays a tumor-suppressive role in most cancers [106], but may also exert context-dependent protumorigenic functions. In melanoma cells under hypoxic conditions, it may promote cell migration and metastasis through PKA-mediated phosphorylation [107].

The results of studies investigating the role of TROP2 in EMT are inconclusive. Numerous findings show that upregulation of TROP2 is associated with EMT induction in various tumor cell types, as evidenced mainly by decreased E-cadherin levels and increased vimentin expression [108,109,110,111], while some authors reported the opposite effect [112]. Our observations agree well with the work of Guerra and colleagues that E-cadherin expression is maintained in TROP2-positive cancer cells and that the expression of transcriptional master regulators that induce EMT does not change significantly [93]. While the authors did not report an upregulation of vimentin in colon cancer cells growing as xenografts in mice but in the stroma, we observed the epithelial induction of vimentin in TROP2-positive cells even at early stages of colon tumorigenesis.

We then focused on the mechanism by which *TROP2* gene expression is “turned on”. First, we analyzed the role of Wnt/β-catenin signaling. Somewhat surprisingly, the results of the luciferase assays show that the promoter region of the *TROP2* gene is probably not activated by the exogenous Wnt3a ligand, but rather by BIO or by stable β-catenin protein The latter two methods aberrantly activate Wnt signaling at the level of the β-catenin destruction complex. In contrast, the *TROP2* promoter was activated by the transcriptional coactivator YAP. The link between the Wnt and Hippo signaling pathways has been previously described at multiple levels [113]. Notably, the transcription factors YAP/TAZ are involved in the β-catenin destruction complex and modulate the transcriptional program of the Wnt pathway. In addition, the transcriptional coactivator YAP is regulated by β-catenin/TCF4 complexes [114]. Interestingly, the silencing of β-catenin further enhanced YAP-dependent transcriptional activation, indicating a possible interference between β-catenin and YAP. We can only speculate about the nature of this interference. Since, in most lineages derived from CRC, a number of genes are regulated by YAP-β-catenin-T-box 5 (TBX5) complexes [115], there is a possibility that a decrease in β-catenin levels increases the amount of free YAP protein in the cell, which in turn leads to an increased expression of genes regulated by this transcriptional regulator in complex with TEAD factors. Another possibility is the transcriptional (co)regulation of *TROP2* by β-catenin and YAP or other transcription factors. A reduction of β-catenin protein levels by RNA interference could allow the regulatory regions of the *TROP2* gene to be occupied more efficiently by these factors, and if they are more potent transcriptional activators than β-catenin-TCF/LEF complexes, there could be an increase in *TROP2* gene transcription. The immunohistochemical staining of neoplastic lesions in mice showed relatively good agreement between TROP2 production and positivity for nuclear, i.e., transcriptionally active, YAP. However, it should be noted here that because of the relatively weak YAP signal localized in the nuclear speckles, it was not possible to accurately determine the extent of overlap between YAP and TROP2 expression. In human adenocarcinomas, the situation was even more unclear, mainly because of the different intensity of staining and the distribution of YAP between the nucleus and cytoplasm. In any case, it is evident that other/additional mechanisms are probably involved in the regulation of TROP2 expression in advanced human tumors. 

It is unclear which signal activates YAP/TAZ signaling or which signal turns off the Hippo pathway, resulting in the transfer of YAP/TAZ regulators to the nucleus. *Trop2* gene expression is specific for epithelial regeneration after an injury, and the transcriptional coactivator YAP is critical for regeneration. Injury and associated tissue reorganization appear to cause the nuclear localization of YAP and the subsequent expression of YAP-regulated genes [116]. In a recent paper, Guillermin and colleagues demonstrated a positive (feed forward) relationship between Wnt/β-catenin signaling and YAP in organoid cells [81]. Briefly, the Wnt signaling pathway activates the transcription of the DNA-binding proteins TEAD (namely TEAD2/4) and YAP, which activate the expression of target genes of the heterocomplex YAP–TEAD in appropriately (externally) stimulated cells. In addition, several studies suggest that the activation of YAP/TAZ signaling in intestinal organoids can be induced by specific culture conditions, in particular, by the prolonged cultivation of organoids in Matrigel or using collagen or other mechanically stiff matrices [116,117,118]. The analysis of organoids led to the following conclusions. Organoids derived from tumors are so heterogeneous that it is difficult to draw a conclusion from comparing the expression of a few selected genes. In organoids derived from healthy tissue, we consistently observed an increased expression of target genes of the Wnt pathway and *TROP2*. The expression was dependent on the degree of stimulation of Wnt signaling. The situation was less clear for the potential target genes of the YAP/TAZ signaling, but this can be explained by the fact that these target genes were identified in a different cellular context, i.e., in organoids derived from embryonic intestinal tissue or in organoids growing under conditions mimicking a damaged intestine. The group of target genes of the YAP/TAZ regulators in “healthy” organoids is much less characterized in contrast to the Wnt pathway. However, it should be noted here that for six of the eight genes tested, i.e., the potential target genes of YAP/TAZ signaling, we found a positive correlation between their expression levels and Wnt-signaling activity. Another possible interpretation for the increased expression of TROP2 after the enhanced stimulation of the Wnt pathway is that this stimulation increases the total percentage of IESCs in organoids. However, because TROP2 expression is undetectable in healthy intestinal stem cells, this possibility seems unlikely.

Another point of discussion is whether TROP2 can be considered “only” a marker for cancer cells in a specific epigenetic state (whether this is the result of extracellular stimuli or internal cellular settings) or whether this protein has a specific functional significance for the cell. In our analysis of TROP2 KO/iTROP2 SW480 cells, we observed a statistically significant but relatively small inhibitory effect on cell migration, but no significant effect on the change in cell proliferation in vitro. Although a trend toward the formation of smaller xenograft tumors was observed in vivo, this difference did not reach statistical significance. We tested the invasiveness of TROP2 KO cells using Matrigel drops and collagen spheroids and found a trend toward a slightly increased invasiveness of the TROP2-positive cell clones, but the results were not reproducible (data not shown). It should also be mentioned here that Trop2-deficient mice are viable and have no obvious developmental defects. However, Trop2^−/−^ keratinocytes transformed with the Hras oncogene preferentially underwent EMT, showed increased proliferative and migratory capacity, and formed tumors with a spindle cell histology [112]. Finally, when analyzing the function of TROP2, we should not forget its possible redundancy with respect to the EPCAM paralog [27].

The epithelial–mesenchymal transition and reverse mesenchymal-epithelial transition (MET) are thought to be essential processes that are critical for the ability of cancer cells to invade surrounding tissues and blood vessels and form distant metastases. Recently, however, it has been demonstrated that cells that are in a hybrid epithelial/mesenchymal (E/M) state, as opposed to fully mesenchymal cells, have an increased potential to form metastases. This state has been shown to provide the greatest capacity for tumor formation [119]. Moreover, the invasiveness in CRC is mostly mediated by cell clusters rather than isolated cells [120]. It seems that the simultaneous presence of epithelial marker E-cadherin and upregulation of mesenchymal proteins (vimentin and fibronectin) may indicate a hybrid E/M state of TROP2-positive cells. Further studies are needed to determine whether TROP2 is a marker for cells that have undergone partial/hybrid EMT or whether it plays a role—and possibly what role it plays—in tumor cell invasiveness and migration.

Do our results suggest the possibility of using the surface expression of TROP2 for the targeted treatment of CRC? At this point, it is difficult to say. If we consider the possibility that TROP2-producing tumor cells have active YAP signaling and that this signaling is important for tumor progression, the answer would be yes. In contrast to the relatively high response rates to the treatment of metastatic TNBC (33.3%) and urothelial carcinoma (28.9%) in the third line of treatment, the response rate to the TROP2-specific Sacituzumab govitecan antibody conjugated to the irinotecan analog SN-38 (trade name TRODELVY) was only 3.2% (95% CI: 0.1–16.7) in the phase 1/2 IMMU-132-01 basket trial involving 31 unselected patients with CRC [121]. This low response rate can be explained by the fact that the vast majority of CRC patients have already been pretreated with irinotecan in previous lines of treatment. It is likely that the response to treatment would have been significantly better if Sacituzumab govitecan had been used in the subgroup with a high TROP2 protein expression in the early stages of treatment or if the anti-TROP2 antibody had been conjugated with another drug.

## 5. Conclusions

In this study, we have shown that TROP2 expression is increased in the early stages of neoplastic transformation of the intestinal epithelium in humans and mice, and that a high TROP2 expression is present in approximately one quarter of the analyzed CRC cases. In CRC cases with a high TROP2 protein positivity, this represents an independent negative prognostic factor and correlates significantly with lymph node metastasis and poor tumor cell differentiation. The whole-genome analysis of the mRNA profiles revealed functionally similar groups of genes associated with TROP2 expression on the cell surface when comparing the cells from neoplastic intestinal lesions in mice or cells from human CRC samples. These genes are associated with EMT, migration/invasiveness, and extracellular matrix interaction/remodeling. The functional analysis of TROP2-deficient CRC cells showed a slightly increased migratory capacity in vitro compared with WT cells. The absence of TROP2 had no effect on cell proliferation in vitro. Furthermore, we demonstrated that the ectopic expression of transcriptional coactivator YAP activated the *TROP2* promoter in the luciferase reporter assay. The possible regulation of TROP2 expression by the active YAP signaling was verified by the knockdown of YAP in CRC cells.

## Figures and Tables

**Figure 1 cancers-14-04137-f001:**
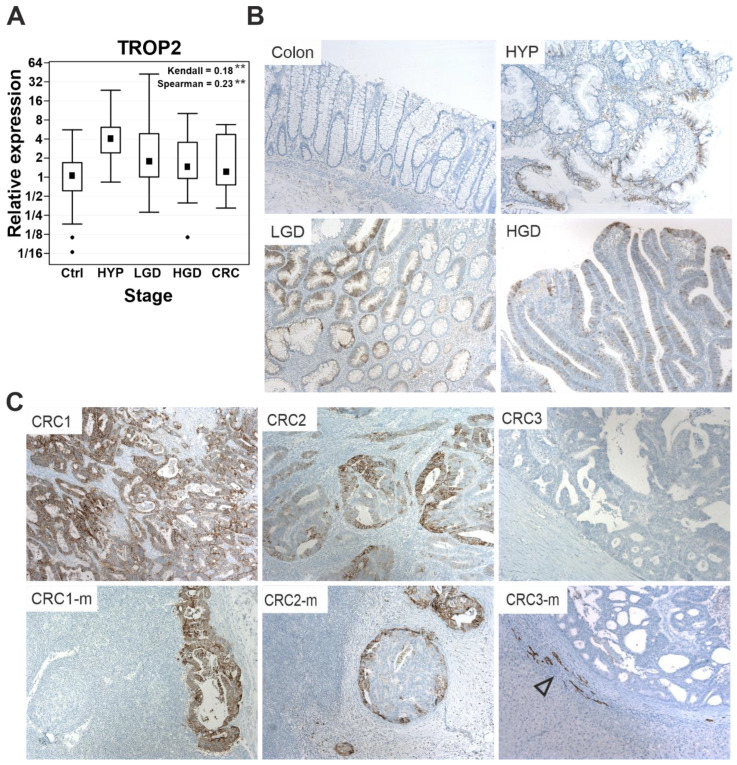
*TROP2* gene expression and protein localization in human adenoma and adenocarcinoma samples. (**A**) Upregulated *TROP2* gene expression at different stages of the neoplastic transformation sequence; *TROP2* expression levels at the indicated tumor stage were compared with healthy colon biopsy samples. Data are presented as medians (black squares), 25th and 75th percentiles (boxed areas), and minimum and maximum values (“whiskers”); outlier values are also indicated (small rotated black squares). The association between the *TROP2* expression profile and the histology grade of the neoplasia is significant, as shown by the Spearman and Kendall coefficient values. **, *p* < 0.01. (**B**) Representative microscopic images of immunohistochemical detection of TROP2 in normal colonic mucosa (colon) and in the indicated types of preneoplastic lesions. (**C**) Heterogeneous TROP2 expression in colorectal cancer; typical staining in adenocarcinomas with strong (CRC1), moderate (CRC2), and no TROP2 expression (CRC3) is shown. The corresponding lymphoid (CRC1-m and CRC2-m) or hepatic (CRC3-m) metastases have a TROP2 staining pattern similar to the primary tumor. Note the positive staining of intrahepatic cholangiocytes (empty arrowhead). All specimens were stained with substrate 3,3-diaminobenzidine (DAB; dark brown precipitate) and counterstained with hematoxylin (blue nuclear stain). Ctrl, healthy colon; HYP, hyperplastic polyps (*n* = 9); LGD, adenomas with low-grade dysplasia (*n* = 24); HGD, adenomas with high-grade dysplasia (*n* = 25); CRC, invasive carcinoma (*n* = 12); original magnification 100×.

**Figure 2 cancers-14-04137-f002:**
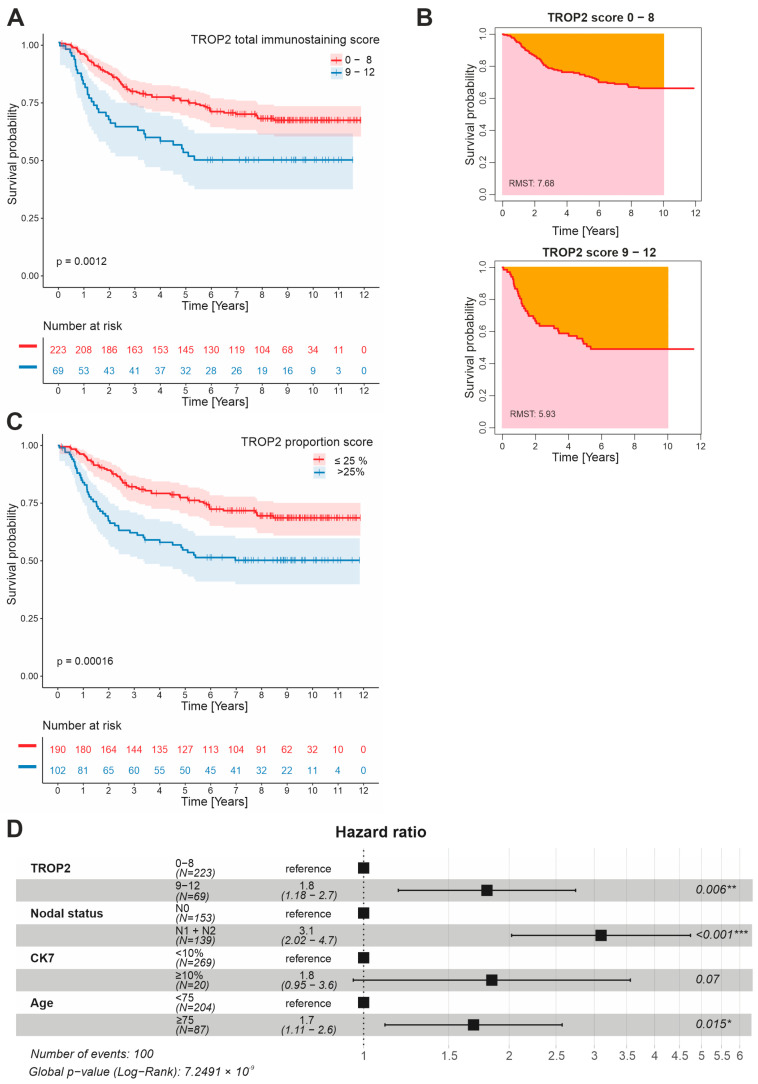
Analysis of cancer-specific survival and prognostic significance of TROP2 in 292 patients with colorectal cancer. (**A**) Kaplan–Meier survival curves calculated for groups with high (score 9–12) and low/medium (score 0–8) TROP2 expression showed poorer survival in patients with TROP2 overexpression. (**B**) Restricted mean survival time (RMST) is significantly shorter in the high TROP2 expression group than in the low/medium score group (5.93 vs. 7.68 years, *p* = 0.0012). (**C**) Kaplan–Meier analysis based on the TROP2 proportion score showed poor prognosis in patients with ≥25% positively stained TROP2 cancer cells (*p* = 0.00016). (**D**) Forrest plot showing significant results of multivariate Cox regression identifying high TROP2 expression as an independent negative prognostic factor in patients with CRC (*p* = 0.006) along with lymph node involvement (*p* < 0.001) and age ≥ 75 years (*p* = 0.015); CK7—cytokeratin 7; * *p* < 0.05; ** *p* < 0.01; *** *p* < 0.001.

**Figure 3 cancers-14-04137-f003:**
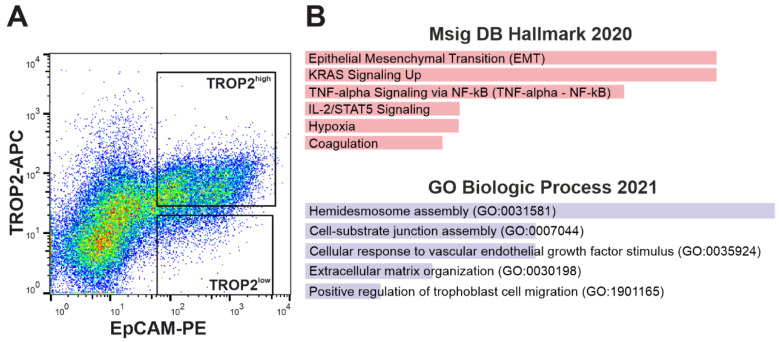
Analysis of human cells obtained from CRC specimens. (**A**) A representative diagram shows the fluorescence-activated cell-sorting gate used to obtain TROP2^high^ and TROP2^low^ human colon tumor cells; see Materials and Methods for further details. (**B**) Analysis of gene expression profiles of TROP2^high^ human tumor cells. Differentially expressed genes (*n* = 72) enriched in TROP2^high^ cells (significance criterion: adjusted *p*-value < 0.05) were analyzed using the Molecular Signatures Database (MSigDB) Hallmark 2020 and Gene Ontology (GO) Biological Processes 2021 gene set collections. The adjusted *p*-value (calculated from Fisher’s exact test) was assigned to each category; the top five categories with the adjusted *p*-value < 0.01 are shown. The lengths of the columns correspond to the significance of each column in the graph—the longer the column, the higher the significance. The genes are listed in Appendix A.

**Figure 4 cancers-14-04137-f004:**
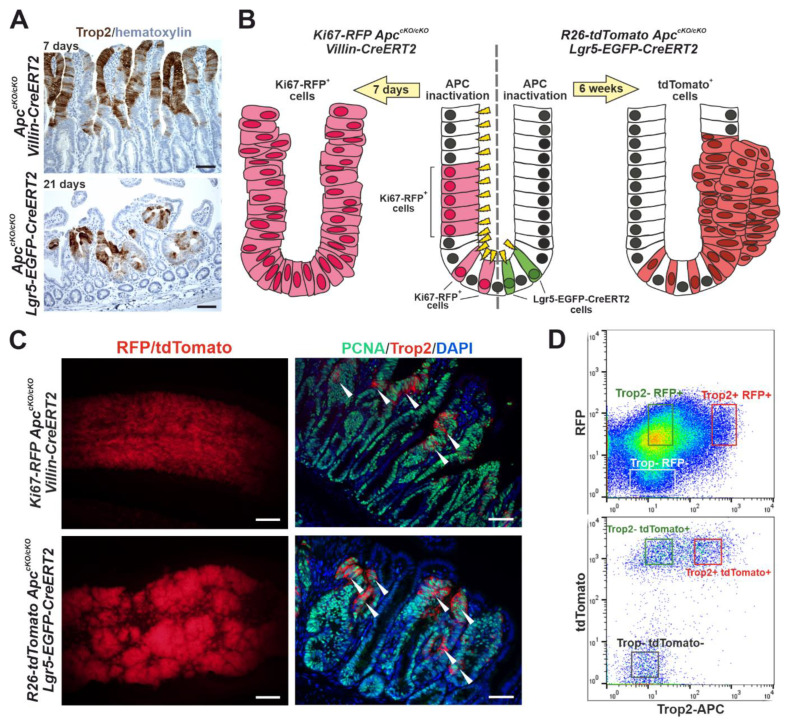
Analysis of Trop2 expression in two mouse models of intestinal tumorigenesis. (**A**) Immunohistochemical detection of Trop2 in the indicated mouse strains 7 (**upper** image) or 21 (**bottom** image) days after tamoxifen administration. Specimens were counterstained by hematoxylin; scale bar: 100 µm. (**B**) Schematic diagram of the experimental setup for tumor cell isolation. The middle part of the diagram shows a crypt with the dividing cells (these cells produce the Ki67-RFP fusion protein) and stem cells expressing EGFP and CreERT2 recombinase; cells containing active (nuclear) Cre recombinase (after tamoxifen administration) are labeled by yellow arrowheads. The left and right parts of the diagram depict neoplastic intestinal lesions (marked by red fluorescence) that developed after Apc inactivation with the indicated Cre driver. (**C**) Stereomicroscopic images of native RFP/tdTomato fluorescence (**left**) and immunodetection of PCNA- and Trop2-positive (white arrowheads; **right**) cells in the middle part of the small intestine 7 days (**top** image) and 6 weeks (**bottom** image) after Apc inactivation. The specimens were counterstained with 4′,6-diamidine-2′-phenylindole dihydrochloride (DAPI; nuclear blue florescent signal). (**D**) Sorting of cell populations used for expression profiling. Scale bars: 2 mm (**left** panel) and 100 µm (**middle** panel).

**Figure 5 cancers-14-04137-f005:**
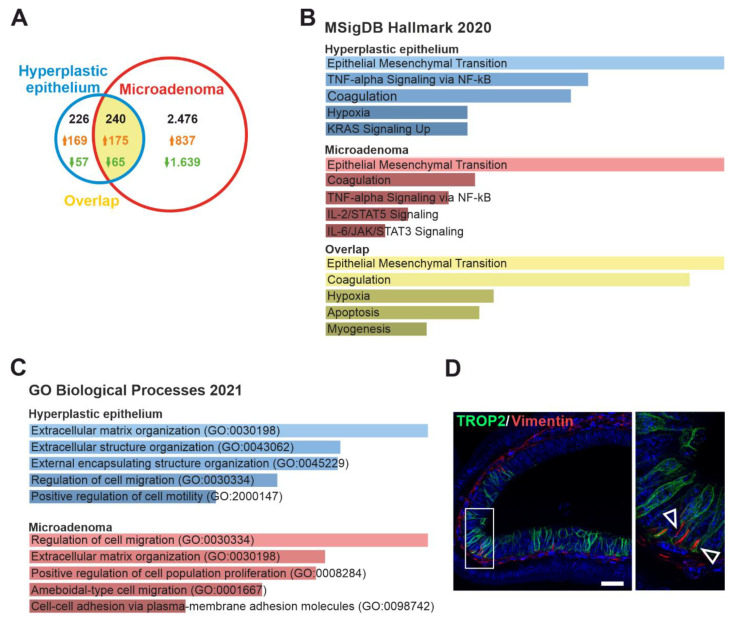
Analysis of gene expression profiles of Trop2^+^ tumor cells isolated from Apc-deficient neoplasia developed in two mouse models of *Apc* gene inactivation. (**A**) Venn diagram indicating the number of genes differentially expressed in Trop2-positive cells (compared to Trop2-negative cells) isolated from the hyperplastic epithelium of *Ki67-RFP Apc^cKO/cKO^ VillinCreERT2* mice or microadenomas of *Rosa26-tdTomato Apc^cKO/cKO^ Lgr5-EGFP-CreERT2* mice 7 days or 6 weeks after Apc inactivation, respectively. Significance criterion: |log_2_ FC| > 1; adjusted *p*-value < 0.01. The genes are listed in Appendix A. Differentially expressed genes were further analyzed using the MSigDB Hallmark 2020 (**B**) and GO Biological Processes 2021 (**C**) gene set collections. The adjusted *p*-value (calculated from Fisher’s exact test) was assigned to each category; five top categories with the adjusted *p*-value < 0.01 are shown. Note that the statistical significance was not reached for the overlap gene set using when the GO Biological Processes 2021 database was used. The coloring and length of the columns corresponds to the significance of each column in the graph—the longer column and lighter color indicate higher significance. (**D**) Immunodetection of Trop2 and vimentin in microadenomas that developed in *Apc^cKO/cKO^ Lgr5-EGFP-CreERT2* mice 6 weeks after Apc inactivation; magnified image of the inset is on the right. Arrowheads point to Trop2-positive epithelial cells with accumulated vimentin. Specimens were counterstained with DAPI; scale bar: 50 μm.

**Figure 6 cancers-14-04137-f006:**
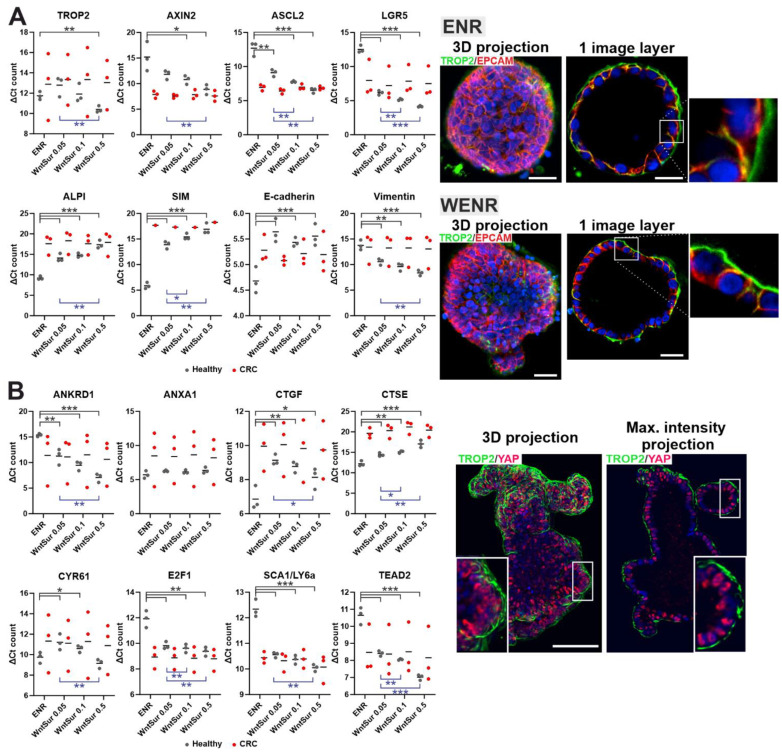
Hyperactivation of the Wnt pathway stimulates TROP2 in healthy colon organoids. (**A**) Left, qRT-PCR analysis of expression levels of the indicated genes in human healthy colon organoids (Healthy) and organoids derived from human CRC. The organoids were cultured either in standard organoid medium (ENR), or in ENR medium supplemented with next-generation surrogate Wnt ligand (WntSur) at final concentrations of 0.05 nM, 0.1 nM, and 0.5 nM (WntSur 0.05, WntSur 0.1, and WntSur 0.5, respectively). RNA samples isolated from three healthy and three tumor organoid cultures were analyzed (each in technical triplicate; the average value of each triplicate is represented by a grey or red dot, respectively). The diagram shows ΔCt counts, i.e., the cycle threshold (Ct) value of the gene of interest minus the Ct-value of housekeeping gene β-ACTIN (the lower the ΔCt value, the higher the gene expression); black lines indicate the mean value for three biological replicates; * *p* < 0.05; ** *p* < 0.01; *** *p* < 0.001. Wnt/β-catenin target genes—axis inhibition protein 2 (AXIN2), achaete-scute complex homolog 2 (*ASCL2*), leucine-rich repeat containing G protein-coupled receptor 5 (*LGR5*); differentiation markers—alkaline phosphatase (*ALPI*), sucrase-isomaltase (*SIM*); and genes related to the EMT status—E-cadherin and vimentin. Right: fluorescent microscopy images of TROP2 (membranous green fluorescent signal) in tumor organoids derived from human CRC; organoids with intermediate levels of *TROP2* mRNA (ΔCt value ± 13, see the very left diagram) were used for the staining. The organoids were cultured either in the ENR medium alone or in the ENR medium containing 0.5 nM WntSur (WENR). Note that the TROP2 protein expression is independent of the presence of WntSur in the culture medium. The specimen was counterstained with DAPI. Left images show three-dimensional (3D) projection of the whole organoid; right images show one image layer; representative pictures are shown. Scale bar: 10 µm. (**B**) Left: quantitative RT-PCR analysis of expression levels of the YAP/TAZ signaling target genes. The experimental setup is identical to panel (**A**); ANKRD1, ankyrin repeat domain 1; ANXA1, annexin A1; CTGF, connective tissue growth factor; CTSE, cathepsin E; CYR61, cysteine-rich angiogenic inducer 61; E2F1, E2F transcription factor 1; SCA1/LY6a, stem cell antigen-1/lymphocyte antigen 6; TEAD2, TEA domain transcription factor 2. Right: fluorescent microscopy images of TROP2 (membranous green fluorescent signal) and YAP (nuclear red florescent signal) protein localization in a healthy human colon organoid. The organoids were cultured in the WENR medium containing 0.5 nM WntSur. The specimen was counterstained with DAPI. Setting of the left image is described in panel (**A**); right image shows maximal intensity projection of six adjacent layers; the enlarged image is shown in the inset. Scale bar: 25 µm.

**Figure 7 cancers-14-04137-f007:**
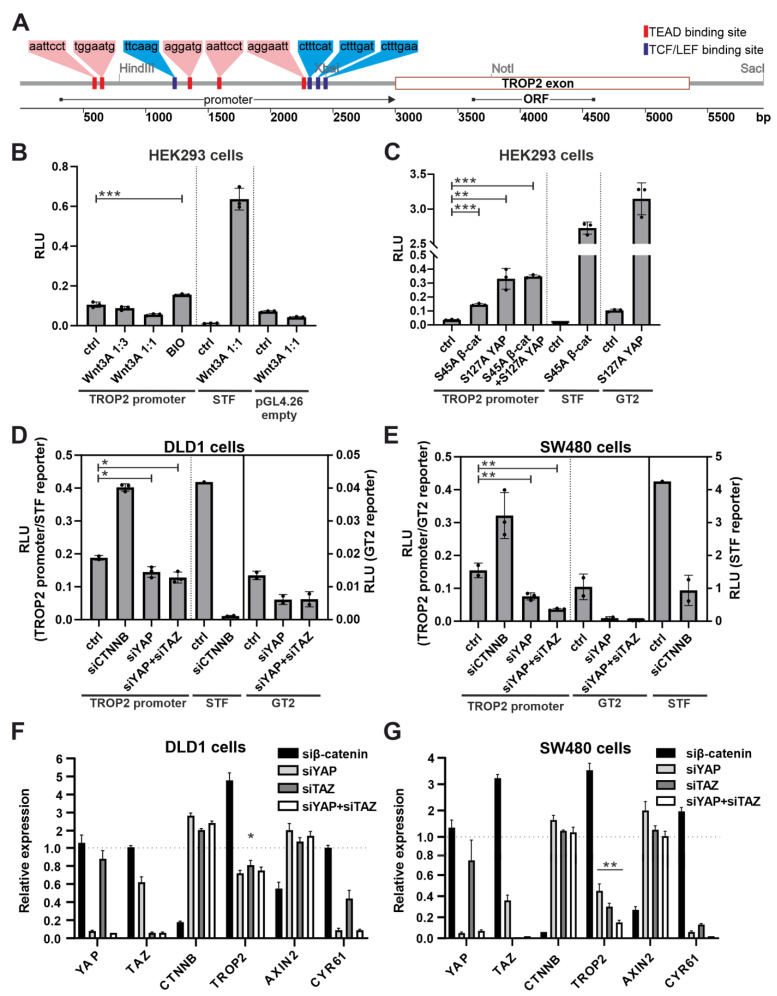
The human *TROP2* promoter harbors multiple binding sites for transcription factors TEAD and TCF/LEF and is activated by YAP. (**A**) A graphical schema of the human *TROP2* locus showing the regulatory/promoter region harboring the TEAD and TCF/LEF binding sites (red and blue bars indicate the relative positions of the above consensus binding sequences within the promoter region) detected by the CiiDER in silico analysis tool. A sequence labeled “promoter” was cloned into the firefly luciferase reporter for functional analysis; ORF represents the open reading frame region of the *TROP2* single-exon gene. The positions of the selected restriction endonuclease sites are also indicated. (**B**) Luciferase assay in HEK293 cells transfected with the pGL4.26 reporter vector containing the *TROP2* promoter region or the Wnt/β-catenin-responsive SuperTopFlash reporter (STF) or the empty pGL4.26 vector. To stimulate the Wnt pathway, transfected cells were treated with increasing concentrations of Wnt3A-conditioned medium (mixed with standard culture medium as indicated) or with GSK3 inhibitor BIO; “ctrl” indicates cells without Wnt3a or BIO treatment. (**C**) Luciferase assay in HEK293 cells co-transfected with the indicated reporters and constructs expressing mutant variants of β-catenin (S45A) or YAP (S127A); “ctrl” indicates cells transfected with the reporter and an empty expression vector. Luciferase reporter assays after knockdown of β-catenin, YAP, and/or TAZ in DLD1 (**D**) and SW480 (**E**) cells. Cells were transfected with the reporter plasmid along with siRNA(s) as indicated. All experiments were performed in triplicate; firefly luciferase levels were normalized to Renilla luciferase levels. Results are expressed as relative luciferase units (RLU). Quantitative RT-PCR analysis of DLD1 (**F**) and SW480 (**G**) cells after knockout of β-catenin, YAP, and/or TAZ. Cells were treated with siRNAs as indicated; results were normalized to *TBP* mRNA levels. The experiment was performed in triplicate, and results are shown as relative expression compared with a sample treated with a non-silencing siRNA. Error bars in all graphs indicate SDs; * *p* < 0.05; ** *p* < 0.01; *** *p* < 0.001.

**Figure 8 cancers-14-04137-f008:**
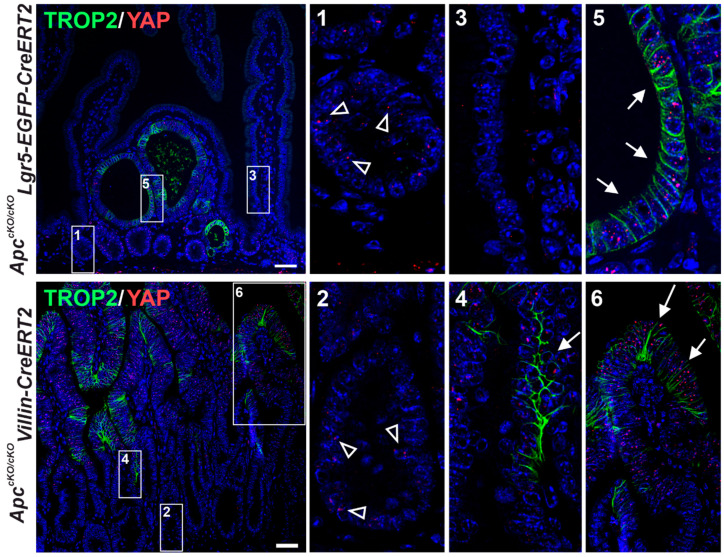
Immunodetection of Trop2 and YAP in the small intestine of the indicated mouse strains 6 weeks (top images) and 7 days (bottom images) after Apc inactivation. Images in insets are magnified on the right; samples were counterstained with DAPI. Arrowheads in the magnified images (1) and (2) indicate nuclei in WT crypts that are sparsely positive for YAP, whereas healthy epithelium on the villus is predominantly YAP negative (3). White arrows mark areas of transformed epithelium at the villus edge of the crypt (4), in adenomas (5), or at the villus tip (6) with Trop2 expression and YAP nuclear localization; scale bar: 50 μm.

**Figure 9 cancers-14-04137-f009:**
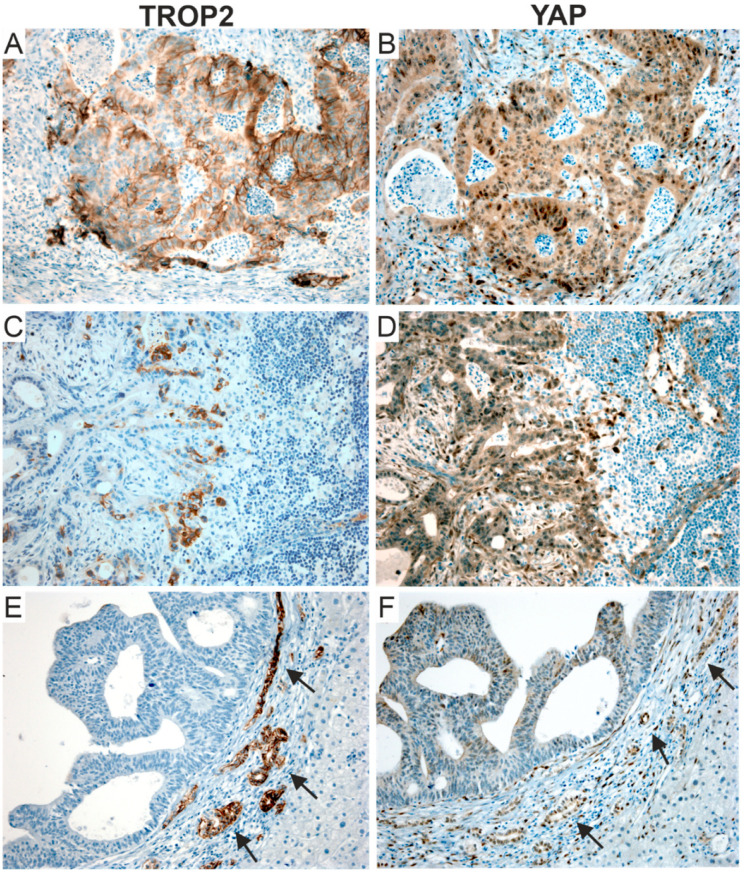
Immunohistochemical detection of TROP2 and YAP in CRCs shows considerable heterogeneity, with YAP showing stronger expression. Serial sections of primary CRC (**A**,**B**) and lymph node (**C**,**D**) and liver metastases (**E**,**F**) were stained for TROP2 and YAP. Positive staining of intrahepatic cholangiocytes (black arrows in **E**,**F**) was used as an internal control; original magnification 200×.

## Data Availability

Minimum Information About a Microarray Experiment (MIAME) compliant data were deposited to the ArrayExpress database under accession numbers: E-MTAB-11377 (analysis of *Ki67-RFP Apc^cKO/cKO^ Villin-CreERT2* mice), E-MTAB-11382 (*Lgr5-EGFP-IRES-CreERT2* mice), and E-MTAB-11466 (human specimens).

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
