# Peer review of "TROP2 Represents a Negative Prognostic Factor in Colorectal Adenocarcinoma and Its Expression Is Associated with Features of Epithelial–Mesenchymal Transition and Invasiveness"

_cancers, 2022, doi:10.3390/cancers14174137_

Round 1
Reviewer 1 Report
The revised version of the manuscript by Švec et al. addressed all of my previous comments. In addition, I only have a few minor comments:
- Line 123 – ‘valine l286’ should be valine 286
- Lines 527-529 – please clarify ‘Principal component analysis (PCA) based on the 500 most differentially expressed genes revealed substantial similarity between TROP2high and TROP2low cells obtained from the same sample, indicating a high degree of heterogeneity between tumors’.
- Line 916 – ‘we found a significant association of high TROP2 protein production […] with poor differentiation (grade 3) and mucinous and signet ring tumor morphology’. These observations should be included in the Results section as well.
- Line 1125 – in the Conclusions section, the phrase ‘Based on these observations, we believe that targeted anti-TROP2 therapy may be a promising approach, especially for the subset of CRC with high TROP2 levels’ is hazardous; previous observations do not correlate in any way with the efficacy of such treatment. I think this should be removed.
Author Response
Reviewer 1:
The revised version of the manuscript by Švec et al. addressed all of my previous comments. In addition, I only have a few minor comments:
- Line 123 – ‘valine l286’ should be valine 286
Answer: We corrected the error; we thank the reviewer for his care and attention (Line 123; the line numbering corresponds to the text format with the "Track changes" function activated and the "All Markup" format).
- Lines 527-529 – please clarify ‘Principal component analysis (PCA) based on the 500 most differentially expressed genes revealed substantial similarity between TROP2high and TROP2low cells obtained from the same sample, indicating a high degree of heterogeneity between tumors’.
Answer: We agree and have reworded the relevant paragraph (Line 531-535).
- Line 916 – ‘we found a significant association of high TROP2 protein production […] with poor differentiation (grade 3) and mucinous and signet ring tumor morphology’. These observations should be included in the Results section as well.
Answer: We agree and have included these observations in the results (Line 496-500).
- Line 1125 – in the Conclusions section, the phrase ‘Based on these observations, we believe that targeted anti-TROP2 therapy may be a promising approach, especially for the subset of CRC with high TROP2 levels’ is hazardous; previous observations do not correlate in any way with the efficacy of such treatment. I think this should be removed.
Answer: We agree, we have removed the sentence.

Reviewer 2 Report
The authors have studied the expression and regulation of the plasma membrane protein TROP2 in intestinal epithelium and cancer. TROP2 is upregulated in a variety of human cancers, which correlates with clinical data, and is a target of antibody-based therapy. The authors show here that TROP2 is upregulated in colorectal cancer, correlating with reduced patient survival. Upregulation was also observed in animal models of intestinal tumorigensis and in tumor organoids revealing a heterogenous pattern of TROP2 protein expression between cells. Upregulation might be a result of Wnt and YAP/Taz signalling, which in part is due to regulation of the TROP2 promoter level as revealed by reporter assays.
Data on the TROP2 expression in colorectal cancer and clinical correlations are partly confirmatory as discussed by the authors but were performed at a high technical level, allowing clear-cut conclusions. The analysis of TROP2 in APC k.o.-based animal models is highly original and convincing, as is the gene expression analysis of TROP2-sorted cancer cells. In addition, the finding that TROp2 might be regulated by YAP/TAZ is novel and interesting.
Altogether, this paper describes a nice and conclusive story with high quality figures.
There are only a few minor comments that should be addressed:
1 There is a discrepancy between text and figure with respect to Fig. 6B. It is stated in the text that the CTSE gene showed no response and that the ANXA1 gene showed a decrease in expression, but in the data of Fig. 6B it is the other way round. Please check and correct.
2 Line 572: Sentence starting ”We knocked out…” is misleading because the term “and then” suggests that mice from the Villin-Cre k.o. were subsequently subjected to crossings with Lgr5-Cre, which becomes clear only later.
3 Fig. 10 is a bit problematic because the changes in cell migration after TROP2 induction are rather minor. Moreover, in the discussion we learn that there was clonal variation between the TROP2 k.o.s and that reconstitution of TROP2 to endogenous levels might be incomplete. Thus, the message form this figure is not clear and conclusive, and it should therefore be deleted. Its content might be combined with Supplementary Figure S6 without discussing it too broadly in the main text.
4 Discussion might be shortened because it sometimes reads more like a review. In particular, the chapter on the functional role of TROP2 (p. 30) seems to be redundant with the Introduction, and results by the authors (Fig. 10, see above) do not add much to this question.
Author Response
Reviewer 2:
The authors have studied the expression and regulation of the plasma membrane protein TROP2 in intestinal epithelium and cancer. TROP2 is upregulated in a variety of human cancers, which correlates with clinical data, and is a target of antibody-based therapy. The authors show here that TROP2 is upregulated in colorectal cancer, correlating with reduced patient survival. Upregulation was also observed in animal models of intestinal tumorigensis and in tumor organoids revealing a heterogenous pattern of TROP2 protein expression between cells. Upregulation might be a result of Wnt and YAP/Taz signalling, which in part is due to regulation of the TROP2 promoter level as revealed by reporter assays.
Data on the TROP2 expression in colorectal cancer and clinical correlations are partly confirmatory as discussed by the authors but were performed at a high technical level, allowing clear-cut conclusions. The analysis of TROP2 in APC k.o.-based animal models is highly original and convincing, as is the gene expression analysis of TROP2-sorted cancer cells. In addition, the finding that TROp2 might be regulated by YAP/TAZ is novel and interesting.
Altogether, this paper describes a nice and conclusive story with high quality figures.
There are only a few minor comments that should be addressed:
1 There is a discrepancy between text and figure with respect to Fig. 6B. It is stated in the text that the CTSE gene showed no response and that the ANXA1 gene showed a decrease in expression, but in the data of Fig. 6B it is the other way round. Please check and correct.
Answer: Yes, the names of these genes were swapped; we have corrected the corresponding place in the text (Line 762-763; the line numbering corresponds to the text format with the "Track changes" function activated and the "All Markup" format), and thank you for your careful correction.
2 Line 572: Sentence starting ”We knocked out…” is misleading because the term “and then” suggests that mice from the Villin-Cre k.o. were subsequently subjected to crossings with Lgr5-Cre, which becomes clear only later.
Answer: We agree, the wording of the paragraph has been corrected (Line 581-583).
3 Fig. 10 is a bit problematic because the changes in cell migration after TROP2 induction are rather minor. Moreover, in the discussion we learn that there was clonal variation between the TROP2 k.o.s and that reconstitution of TROP2 to endogenous levels might be incomplete. Thus, the message form this figure is not clear and conclusive, and it should therefore be deleted. Its content might be combined with Supplementary Figure S6 without discussing it too broadly in the main text.
Answer: Since most reviewers requested a change in the content of Figure 10, we decided, at the suggestion of this reviewer, to remove Figure 10 from the main text of the manuscript and move it to the Supplement (Supplementary Figure S7). Subsequently, we significantly shortened the text in the Results section (Line 866-876) and in the Discussion (Line 1077-1095) with respect to the results (content) presented in this figure.
4 Discussion might be shortened because it sometimes reads more like a review. In particular, the chapter on the functional role of TROP2 (p. 30) seems to be redundant with the Introduction, and results by the authors (Fig. 10, see above) do not add much to this question.
Answer: We admit that the length of the manuscript has increased considerably during the revision process – this was mainly due to the requirements of one of the reviewers. Based on the above recommendation, we have shortened portions of the Discussion, particularly those related to the description of possible TROP2 functions (Line 977-982, 987-995, and 1058-1076}.

This manuscript is a resubmission of an earlier submission. The following is a list of the peer review reports and author responses from that submission.
Round 1
Reviewer 1 Report
The authors present an investigation on TROP2 transcriptional upregulation by YAP in colon cancer cells.
Simple Summary and following sections: the negative prognostic impact of Trop-2 on colon cancer has already been shown, as indicated later in the discussion. Relevant literature should be cited and supporting findings should be cited as such.
Abstract and following sections (e.g. Line 842): “we show that TROP2 is expressed in preneoplastic lesions and its expression is maintained in most colorectal carcinomas (CRCs).”
This has already been shown. Relevant literature should be cited and supporting findings should be cited as such.
Introduction: There is pivotal literature on TROP2 cloning, structure determination, structure/function relationship, role in tumor growth and metastasis that has been essentially ignored. Primary findings, rather than reviews, should be cited in all relevant article sections.
Line 84: “TROP2 is a potential marker for adult stem/progenitor cells.”
- Reference to pivotal literature on the expression and driver role of TROP2 on stem and progenitor cells is largely missing. This should be implemented.
Line 100: Among the wrong statements, is the indication that TROP2 interacts with Cyclin D1. This stems from a gross misunderstanding of TROP2 generating a fusion mRNA with Cyclin D1 (Guerra et al. Cancer Res 2008;68:8113-21). Nothing to do with the protein. This should be stated.
Line 104 and following: “TROP2 activity is modulated by regulated intramembrane proteolysis”.
- TROP2-activatory signaling cascade is triggered by extracellular thryroglobulin domain cleavage. This then activates a proteolytic cascade. Corresponding findings and literature should be presented and discussed.
Line 519, 869 and other sections “Corresponding genes involved in EMT and extracellular matrix reorganization that were upregulated in TROP2-positive cells ..”
- The relationship between TROP2 expression and EMT in colon cancer has been previously studied, and opposite findings were obtained. Such controversy needs to be highlighted and discussed. Database gene categorization should not be used as gene function classification criterium.
Line 722 “In healthy organoids, we found that increased stimulation of the Wnt pathway resulted mainly in increased expression of target genes of YAP/TAZ, although for some genes we observed the opposite trend, i.e., a decrease in their expression after stimulation by the WntSur ligand (e.g., for the CTSE gene), or they did not respond to this type of stimulation (e.g., for the ANXA1 gene).”
And
Line 767 “Overall, no to a moderate response of the TROP2 promoter to exogenous stimulation by Wnt3a or treatment with BIO was detected”
- These findings show no causal relationship to Wnt, and this should be clearly stated. See also the Discussion.
Line 772, 896 and other sections “knockdown of CTNNB mRNA increased luciferase expression of the TROP2 promoter, whereas YAP or simultaneous knockdown of YAP and TAZ decreased TROP2 promoter activity in DLD1 and more significantly in SW480 cells”
- These findings do not prove TROP2 transcriptional activation by YAP/TAZ. If anything, they indicate the opposite.
Line 783, 907 and other sections “investigated whether colocalization of TROP2 and YAP also occurs in human tumors by analyzing selected CRC cases (Figure 9). In contrast to TROP2 immunostaining, cytoplasmic and scattered nuclear expression of YAP was detected in the majority of samples examined. In some cases, nuclear YAP positivity colocalized with strong membranous TROP2 staining (Figure 9A and 9B). In other cases, the overlap was observed only in limited areas of the tumor, including neoplastic cells forming the invasive margin in lymph node metastases (Figure 9C and 789 9D). However, we also observed heterogeneous nuclear YAP positivity that was not followed by detectable TROP2 expression”
- These findings do not prove co-expression of TROP2 and YAP in Colon cancer.
Line 952 “Thus, this positive loop between TROP2, Wnt/β-catenin, and YAP signaling may contribute to the enhanced malignant properties of TROP2-positive tumor cells.”
- No evidence was provided for such positive loop.
Line 1005: “the results of colocalization of TROP2 protein and nuclear YAP in CRC are inconclusive”
- This unfortunately adds to concern on the TROP2/YAP relationship, which appears essentially unsupported.
MINOR POINTS
Line 826 “re-expression of TROP2 reduced cell migration”
- Opposite findings have been previously published. This needs to be discussed and validated.
Figure 10B (right panel)
- The scatter plots with or without Doxycyclin induction are essentially identical. Difficult to see the relationship with the statistical significance P values.
Author Response
Reviewer #1
We thank all reviewers for all useful comments on our manuscript. We have tried to incorporate most of them in the revised version of the manuscript. In addition to the required changes, we have added the results of quantitative RT-PCR in DLD1 and SW480 cells (Figure 10F and 10G), which show (confirm) that knockdown of YAP and TAZ reduces TROP2 mRNA expression. These results suggest possible regulation of TROP2 by YAP/TAZ signaling. We added a large amount of information on the structure, posttranslational processing, function, and expression of TROP2 in different cell systems (these changes are mainly related to the comments of reviewer #1). Considering that this is an original article and not a review article, we believe that the introduction must contain sufficient information to understand the context of the results and discussion. However, because Cancers does not impose a limit on the text length, we have attempted to address most comments and objections, although we have significantly expanded the text and the number of citations for this purpose. We hope that these changes have helped to improve the manuscript. Finally, we would like to emphasize that the main advantage of our study is that we used primary cells isolated from human or mouse tumors to analyze the expression profiles of TROP2-positive cells. This makes our results interesting and represents an important complement to data obtained mainly with cell lines.
Comment: The authors present an investigation on TROP2 transcriptional upregulation by YAP in colon cancer cells.
Simple Summary and following sections: the negative prognostic impact of Trop-2 on colon cancer has already been shown, as indicated later in the discussion. Relevant literature should be cited and supporting findings should be cited as such.
Answer: We have added a more detailed description of the previously published results on the prognostic role of TROP2 in CRC patients, including one new citation (Discussion line 910-920). The fact that TROP2 expression is a negative prognostic factor in colorectal cancer was mentioned in the Simple Summary.
Comment: Abstract and following sections (e.g. Line 842): “we show that TROP2 is expressed in preneoplastic lesions and its expression is maintained in most colorectal carcinomas (CRCs).”
This has already been shown. Relevant literature should be cited and supporting findings should be cited as such.
Answer: We have added a more detailed description of the previously published results on TROP2 staining of a preneoplastic lesion, including two new citations (Discussion line 887-906). We modified the Abstract.
Comment: Introduction: There is pivotal literature on TROP2 cloning, structure determination, structure/function relationship, role in tumor growth and metastasis that has been essentially ignored. Primary findings, rather than reviews, should be cited in all relevant article sections.
Answer: We have added a brief description of the TROP2 structure and its role in tumor progression, including five new citations (Introduction line 77-86).
Comment: Line 84: “TROP2 is a potential marker for adult stem/progenitor cells.”
- Reference to pivotal literature on the expression and driver role of TROP2 on stem and progenitor cells is largely missing. This should be implemented.
Answer: The paragraph describing the relationship between TROP2 and stem cells has been modified and seven additional citations have been added (Introduction line 96-106).
Comment: Line 100: Among the wrong statements, is the indication that TROP2 interacts with Cyclin D1. This stems from a gross misunderstanding of TROP2 generating a fusion mRNA with Cyclin D1 (Guerra et al. Cancer Res 2008;68:8113-21). Nothing to do with the protein. This should be stated.
Answer: Due to the already considerable expansion of the Introduction, we have decided to remove the detailed description of the TROP2-Cyclin D1 chimeric mRNA from the manuscript, as it is not directly related to the results presented, and Cyclin D1 has been removed from the list of TROP2 "interactors."
Comment: Line 104 and following: “TROP2 activity is modulated by regulated intramembrane proteolysis”.
- TROP2-activatory signaling cascade is triggered by extracellular thryroglobulin domain cleavage. This then activates a proteolytic cascade. Corresponding findings and literature should be presented and discussed.
Answer: The paragraph describing proteolytic cleavage of TROP2 has been changed and three additional citations have been added (Introduction line 116-126).
Comment: Line 519, 869 and other sections “Corresponding genes involved in EMT and extracellular matrix reorganization that were upregulated in TROP2-positive cells ..”
- The relationship between TROP2 expression and EMT in colon cancer has been previously studied, and opposite findings were obtained. Such controversy needs to be highlighted and discussed. Database gene categorization should not be used as gene function classification criterium.
Answer: The section discussing the relationship between TROP2 and EMT has been modified and additional citations have been added (Discussion line 966-988).
Comment: Line 722 “In healthy organoids, we found that increased stimulation of the Wnt pathway resulted mainly in increased expression of target genes of YAP/TAZ, although for some genes we observed the opposite trend, i.e., a decrease in their expression after stimulation by the WntSur ligand (e.g., for the CTSE gene), or they did not respond to this type of stimulation (e.g., for the ANXA1 gene).”
And
Line 767 “Overall, no to a moderate response of the TROP2 promoter to exogenous stimulation by Wnt3a or treatment with BIO was detected”
- These findings show no causal relationship to Wnt, and this should be clearly stated. See also the Discussion.
Answer: Indeed, we see slight activation of the TROP2 promoter after the treatment of cells with GSK3 kinase inhibitor BIO and also after expression of a stable (oncogenic) variant of b-catenin. The TROP2 promoter was not stimulated by administration of the culture medium containing the Wnt3a ligand. The latter stimulation appears to be very similar to the "normal" (physiological) stimulation of the Wnt pathway. Thus, the first two cause hyperactivation of the Wnt pathway. We have changed the wording in the appropriate places in the text (Results line 801-818; Discussion line 999-1005, 1027-1041).
Comment: Line 772, 896 and other sections “knockdown of CTNNB mRNA increased luciferase expression of the TROP2 promoter, whereas YAP or simultaneous knockdown of YAP and TAZ decreased TROP2 promoter activity in DLD1 and more significantly in SW480 cells”
- These findings do not prove TROP2 transcriptional activation by YAP/TAZ. If anything, they indicate the opposite.
Answer: This is probably a misinterpretation. Yes, we observed that down-regulation of b-catenin increases TROP2 promoter activity and TROP2 gene expression, and we have discussed this fact in the previous sections. However, reporter experiments with the vector producing the YAP activation variant (S127A YAP) and RNA interference clearly show that TROP2 is regulated by YAP and probably by TAZ.
Comment: Line 783, 907 and other sections “investigated whether colocalization of TROP2 and YAP also occurs in human tumors by analyzing selected CRC cases (Figure 9). In contrast to TROP2 immunostaining, cytoplasmic and scattered nuclear expression of YAP was detected in the majority of samples examined. In some cases, nuclear YAP positivity colocalized with strong membranous TROP2 staining (Figure 9A and 9B). In other cases, the overlap was observed only in limited areas of the tumor, including neoplastic cells forming the invasive margin in lymph node metastases (Figure 9C and 789 9D). However, we also observed heterogeneous nuclear YAP positivity that was not followed by detectable TROP2 expression”
- These findings do not prove co-expression of TROP2 and YAP in Colon cancer.
Answer: We mention in the Discussion the technical difficulties in colocalizing TROP2 and YAP in tumor samples (line 1010-1014). Regulation of TROP2 expression by YAP was observed in vitro in CRC cells, so we refined the appropriate formulations throughout the manuscript.
Comment: Line 952 “Thus, this positive loop between TROP2, Wnt/β-catenin, and YAP signaling may contribute to the enhanced malignant properties of TROP2-positive tumor cells.”
- No evidence was provided for such positive loop.
Answer: In view of the results of the RT-PCR analysis showing a possible interference/crosstalk between the Wnt/b-catenin and YAP /TAZ signaling pathways, we deleted the model of positive interaction between the two pathways (Fig. 11) and deleted the corresponding parts of the text.
Comment: Line 1005: “the results of colocalization of TROP2 protein and nuclear YAP in CRC are inconclusive”
- This unfortunately adds to concern on the TROP2/YAP relationship, which appears essentially unsupported.
Answer: The term used, "inconclusive," is inappropriate. As mentioned in the Discussion (line 1010-1014), the analysis of the expression of YAP is more complicated because it is not only the overall level of expression that is important, but also the intracellular distribution of the YAP protein in the cell. This is difficult to determine in tumor samples. We removed the formulation from Conclusions.
MINOR POINTS
Comment: Line 826 “re-expression of TROP2 reduced cell migration”
- Opposite findings have been previously published. This needs to be discussed and validated.
Answer: The section discussing the effects of TROP2 on cell migration/invasion has been modified and additional citations have been added (Discussion line 1049-1063).
Comment: Figure 10B (right panel)
- The scatter plots with or without Doxycyclin induction are essentially identical. Difficult to see the relationship with the statistical significance P values.
Answer: The layout of the chart was changed, which now contains average values and standard deviations in addition to its own measured values (points), and has the form of a bar chart.
Reviewer 2 Report
In this manuscript, Svec et al investigated the possible role of Trophoblast cell surface antigen (TROP)2 in colorectal cancer (CRC) by analysing its expression at mRNA and protein levels in human preneoplastic lesions, CRC tumours, and organoids obtained from healthy colonic epithelium or tumours. They found that high TROP2 positivity correlates with lymph node metastasis and poor tumour differentiation, indicating it as a negative prognostic factor. They also found that TROP2 expression is associated with the expression of genes involved in epithelial-mesenchymal transition, regulation of cell migration and invasiveness, and remodelling of the extracellular matrix. They also identified Yes1-associated transcriptional regulator (YAP) as a novel activator of TROP2 expression, suggesting a possible positive feedback loop between abnormally activated Wnt/beta-catenin signalling, YAP and TROP2 expression. Overall, this is a very well written manuscript with a large amount of data that is comprehensively described. The present work is not only scientifically sound but also a significant contribution to the field. I have a few minor doubts that are described below.
I1. In Figure 1, an increase in TROP2 expression at mRNA level in LGD, HGD and CRC stages (Figure 1A) seem to be less when compared to that at the protein level (Figure 1B/C). This discrepancy should be briefly described in the discussion section.
22. In Figure 10B, why iTROP2 cells (SW480 cells with TROP2 knockout [KO] expressing TROP2 under doxycycline [DOX] inducible promoter) cultured without DOX show decreased migration compared to SW480 cells with TROP2 KO though both the types of cells lack TROP2 expression? Also, the difference in migration between iTROP2 cells cultured without DOX and with DOX is very small. They need to be discussed in the discussion section.
Author Response
Reviewer #2
We thank all reviewers for all useful comments on our manuscript. We have tried to incorporate most of them in the revised version of the manuscript. In addition to the required changes, we have added the results of quantitative RT-PCR in DLD1 and SW480 cells (Figure 10F and 10G), which show (confirm) that knockdown of YAP and TAZ reduces TROP2 mRNA expression. These results suggest possible regulation of TROP2 by YAP/TAZ signaling. We added a large amount of information on the structure, posttranslational processing, function, and expression of TROP2 in different cell systems (these changes are mainly related to the comments of reviewer #1). Considering that this is an original article and not a review article, we believe that the introduction must contain sufficient information to understand the context of the results and discussion. However, because Cancers does not impose a limit on the text length, we have attempted to address most comments and objections, although we have significantly expanded the text and the number of citations for this purpose. We hope that these changes have helped to improve the manuscript. Finally, we would like to emphasize that the main advantage of our study is that we used primary cells isolated from human or mouse tumors to analyze the expression profiles of TROP2-positive cells. This makes our results interesting and represents an important complement to data obtained mainly with cell lines.
Comments and Suggestions for Authors
In this manuscript, Svec et al investigated the possible role of Trophoblast cell surface antigen (TROP)2 in colorectal cancer (CRC) by analysing its expression at mRNA and protein levels in human preneoplastic lesions, CRC tumours, and organoids obtained from healthy colonic epithelium or tumours. They found that high TROP2 positivity correlates with lymph node metastasis and poor tumour differentiation, indicating it as a negative prognostic factor. They also found that TROP2 expression is associated with the expression of genes involved in epithelial-mesenchymal transition, regulation of cell migration and invasiveness, and remodelling of the extracellular matrix. They also identified Yes1-associated transcriptional regulator (YAP) as a novel activator of TROP2 expression, suggesting a possible positive feedback loop between abnormally activated Wnt/beta-catenin signalling, YAP and TROP2 expression. Overall, this is a very well written manuscript with a large amount of data that is comprehensively described. The present work is not only scientifically sound but also a significant contribution to the field. I have a few minor doubts that are described below.
Comment: I1. In Figure 1, an increase in TROP2 expression at mRNA level in LGD, HGD and CRC stages (Figure 1A) seem to be less when compared to that at the protein level (Figure 1B/C). This discrepancy should be briefly described in the discussion section.
Answer: This is a good point, for which we thank you. We discuss the noted contradiction in lines 892-901.
Comment: 22. In Figure 10B, why iTROP2 cells (SW480 cells with TROP2 knockout [KO] expressing TROP2 under doxycycline [DOX] inducible promoter) cultured without DOX show decreased migration compared to SW480 cells with TROP2 KO though both the types of cells lack TROP2 expression? Also, the difference in migration between iTROP2 cells cultured without DOX and with DOX is very small. They need to be discussed in the discussion section.
Answer: Yes, we agree with both points and we added them to the discussion (line 1067-1077).
Reviewer 3 Report
The manuscript by Švec et al. includes extensive work meant to investigate the potential of targeting TROP2 for the treatment of colorectal cancer (CRC). This work is welcome in the context of an ongoing need to find new prognostic biomarkers for CRC. Data was gathered from a large cohort of patients diagnosed with CRC, as well as a variety of sources, including murine models of intestinal cancer, and healthy controls. The main findings of the article include that TROP2 is overexpressed during tumorigenesis in intestinal epithelium, elevated expression of TROP2 was correlated with decreased patient survival and increased tumor aggressiveness and metastasis in CRC patients, and YAP was identified as a novel activator of TROP2 expression.
All work has been carefully designed and thoroughly discussed. However, there are a few minor issues that need to be addressed before publication:
- I believe that the role of TROP2 in drug resistance should be discussed briefly in the introduction, given this is a theme of interest in current research
- Although I appreciate the honesty in the Conclusions section when the authors state that ‘honestly, it’s hard to say’ whether TROP2 is a suitable target for the treatment of CRC or not, I don’t find it appropriate for a scientific paper, and the tone does not match the rest of the paper; I suggest changing this to something more formal
- The conclusions should be rewritten so as to be more specific to the findings presented in the manuscript; lines 1012-1015 could be shortened or moved to the Discussion section
Author Response
Reviewer #3
We thank all reviewers for all useful comments on our manuscript. We have tried to incorporate most of them in the revised version of the manuscript. In addition to the required changes, we have added the results of quantitative RT-PCR in DLD1 and SW480 cells (Figure 10F and 10G), which show (confirm) that knockdown of YAP and TAZ reduces TROP2 mRNA expression. These results suggest possible regulation of TROP2 by YAP/TAZ signaling. We added a large amount of information on the structure, posttranslational processing, function, and expression of TROP2 in different cell systems (these changes are mainly related to the comments of reviewer #1). Considering that this is an original article and not a review article, we believe that the introduction must contain sufficient information to understand the context of the results and discussion. However, because Cancers does not impose a limit on the text length, we have attempted to address most comments and objections, although we have significantly expanded the text and the number of citations for this purpose. We hope that these changes have helped to improve the manuscript. Finally, we would like to emphasize that the main advantage of our study is that we used primary cells isolated from human or mouse tumors to analyze the expression profiles of TROP2-positive cells. This makes our results interesting and represents an important complement to data obtained mainly with cell lines.
Comments and Suggestions for Authors
The manuscript by Švec et al. includes extensive work meant to investigate the potential of targeting TROP2 for the treatment of colorectal cancer (CRC). This work is welcome in the context of an ongoing need to find new prognostic biomarkers for CRC. Data was gathered from a large cohort of patients diagnosed with CRC, as well as a variety of sources, including murine models of intestinal cancer, and healthy controls. The main findings of the article include that TROP2 is overexpressed during tumorigenesis in intestinal epithelium, elevated expression of TROP2 was correlated with decreased patient survival and increased tumor aggressiveness and metastasis in CRC patients, and YAP was identified as a novel activator of TROP2 expression.
Comment: All work has been carefully designed and thoroughly discussed. However, there are a few minor issues that need to be addressed before publication:
- I believe that the role of TROP2 in drug resistance should be discussed briefly in the introduction, given this is a theme of interest in current research
Answer: Yes, we agree with this point and have added a paragraph in the introduction describing the possible involvement of TROP2 in drug resistance (Introduction line 137-153).
Comment: - Although I appreciate the honesty in the Conclusions section when the authors state that ‘honestly, it’s hard to say’ whether TROP2 is a suitable target for the treatment of CRC or not, I don’t find it appropriate for a scientific paper, and the tone does not match the rest of the paper; I suggest changing this to something more formal
- The conclusions should be rewritten so as to be more specific to the findings presented in the manuscript; lines 1012-1015 could be shortened or moved to the Discussion section
Answer: We admit that we have used a somewhat popular narrative tone at the end of the article. We have changed the conclusions accordingly and moved some parts of them to the end of the Discussion.